# Affective evaluation of images influences personality judgments through gaze perception

**Risako Shirai** [1,2,3] *, **Hirokazu Ogawa** [1]

**1** Department of Integrated Psychological Sciences, Kwansei Gakuin University, Nishinomiya, Japan,
**2** Japan Society for the Promotion of Science, Tokyo, Japan, **3** Faculty of Science and Engineering, Waseda University, Tokyo, Japan

* RisakoShirai@gmail.com

## Abstract

Faces that consistently shifted the gaze to subsequent target locations in a gaze cueing task were chosen as being more trustworthy than faces that always looked away from the target, suggesting that the validity of a gaze cue influenced the viewers' judgments regarding the trustworthiness of human faces. We investigated whether the gaze cueing effect and judgments regarding the personality conveyed by a face would be affected by the valence of a target. A face image moved its eyes to the left or the right, and an emotional target image (positive, negative, or neutral) appeared to left or right of the face. Participants had to indicate the location of this target by pressing a key. The target image was preceded by a face that shifted its gaze to the target image (valid cue), a face that directed its gaze to the opposite side (invalid cue), or a face that did not move its eyes (no cue). The perceived trustworthiness of the face was evaluated after the gaze-cueing task. Results showed that faces that looked at positive targets were evaluated as more trustworthy than faces that looked at negative targets. However, the valence of the targets did not affect trustworthiness ratings in invalid and no-cue conditions. We suggest that integrated information about the predictability of the gaze cue and the valence of the gaze target modulates impressions about the personality of the face.

## Introduction

Since cooperative relationships always come with a risk of betrayal, we as individuals must be able to make judgments of whether or not others can be trusted. Cosmides argued that humans have a domain-specific "cheater detection module," which indicates that we are good at detecting violations of social rules [1]. Indeed, it has been suggested that humans can recognize deceptive faces better than cooperative faces (e.g., [2]).

Various factors can influence the judging of the personality of others. One of these factors involves the kind of emotional information that is associated with a to-be-judged person. Evaluative conditioning (EC) has been used to study how pairing an object or person with

**Data Availability Statement:** All relevant data underlying the findings are within our paper and its Supporting information files.

**Funding:** This work was supported by the Japan Society for the Promotion of Science (JSPS

KAKENHI) (https://www.jsps.go.jp/english/index.
html) in the form of grants awarded to RS
(JP18J12574, JP20J00838) and HO
(JP18K03191). The funders had no role in study
design, data collection and analysis, decision to
publish, or preparation of the manuscript.

**Competing interests:** The authors have declared
that no competing interests exist.

emotional stimuli changes the evaluation of an object or person. For example, Kocsor and
Bereczkei showed that those faces previously paired with positive images were rated as more
trustworthy than other faces paired with negative images [3, 4]. Thus, emotional information
surrounding a face can influence facial perception.

Furthermore, since gaze direction is a powerful cue for inferring the internal mental state of
other people, our cognitive/attentional system has developed a measure for using information
involving gaze direction that facilitates social interaction with others. For example, when we
observe a person looking at a particular location, our attention automatically shifts to the same
location, and as a result, we can share the information with each other ("joint attention"; [5–
7]). As well, recent studies have suggested that gaze perception affects personality impressions
of faces [8–12]. Bayliss and Tipper used stimuli involving a face image in which the eyes were
directed either leftward or rightward; this was then followed by a target object that appeared
either at the left or the right side of the face [8]. A viewer's task was to discriminate, as quickly
as possible, whether the target object was a kitchen tool or a garage tool. Also, the trustworthi-
ness of the cueing face was rated after the response to the target. The results showed that the
faces that consistently looked at the location of the target (a "cooperative face") were evaluated
as more trustworthy than the faces that always looked away from the location of the target (a
"deceptive face"). Bayliss and Tipper suggested that information regarding the validity of a
gaze cue was implicitly encoded and associated with the face identities during a gaze-cueing
task, and this information then modulates a viewer's impression of the personality conveyed
by a face.

Since we can infer from the gaze direction of others what they favor and where their interest
lies in social interactions, it is critically important for us to be able to know the object of
another person's gaze. Thus, it is plausible that not only evaluation of the personality impres-
sion of a gazer but also likability of gazed objects is affected through gaze perception. Several
studies have demonstrated that objects consistently gazed at by the faces were liked more than
the objects ignored by these faces [13, 14]. Furthermore, the trustworthiness of the gazer might
modulate the gaze-mediated liking for objects [10, 12, 15]. Thus, King et al. showed that
viewed objects became more likable when they were gazed at by trustworthy faces than by
untrustworthy faces, suggesting that the positive evaluation of the gazer facilitated gaze-medi-
ated preference for the object [10], which then raises the issue of a converse effect. Does the
emotional valence of a gazed object affect the facial impression of the gazer? It is reasonable to
assume this type of reaction is possible because what a person is interested in can be a clue to
understanding his/her personality. However, to the best of our knowledge, no prior research
has addressed the issue.

The purpose of the present study is to clarify whether the affective evaluation of images,
presented as gaze targets, influences facial trustworthiness through gaze perception. The par-
ticipants engaged in a typical gaze cueing task where the target images were visual scenes that
contained emotionally negative (e.g., snakes) or positive objects (e.g., cakes). After the gaze-
cueing trial, the trustworthiness of the faces was evaluated. Facial trustworthiness was com-
pared among the two types of emotions in the gazed images. We predicted that the emotional
valence of images would affect personality judgments only when the face looked toward the
target rather than away from it. Thus, a face that consistently looks at a positive target image
would be evaluated as representing a more trustworthy personality than a face that consistently
looks at a negative image. By contrast, we did not expect such differences in facial evaluation
when faces consistently looked away from emotional targets. Moreover, previous studies have
demonstrated that the perceived impression of a face was altered by repeated paired presenta-
tions of emotional images (e.g., [4, 5]). Therefore, our results might include the effect of such
simple repeated presentations. To control for this possibility, we also prepared a face without

moving eyes. If the effect of the valence of the gazed image on facial evaluation included the effect of repeated presentations of emotional images, we might expect to see the effect of emotional images even if the eyes on the face did not move. On the other hand, if the effect of repeated presentations did not include the results of the facial evaluation, we expected no effect of the emotional images when the eyes on the face did not move.

## Experiment 1

### Materials and methods

**Participants.** We conducted an advance power analysis (G*power; [16, 17]) in to determine the minimum sample size by adopting the following settings: a medium effect size ($f$) of 0.25, a significance level of 5%, 1 group, 6 measurements, a correction for nonsphericity of 0.2, and an intra-class correlation coefficient of zero. It was decided that a minimum of 18 participants was needed to achieve a power level of 0.80. No previous study had identical experimental design as the current study, and it was difficult to know the appropriate effect size in advance. Therefore, we used the medium effect size in order to avoid a large power overflow or too little power. Moreover, we investigated relatively more participants compared to the calculated sample size after considering the possibility of missing data. Finally, 20 graduate and undergraduate students from Kwansei Gakuin University (3 men and 17 women, mean age = 20.65 years) participated in Experiment 1. Ethical approval for all the experiments in our study was obtained by the Kwansei Gakuin University Institutional Review Board for Behavioral Research with Human Participants. All participants reported having normal vision or corrected-to-normal vision and provided their informed consent for participation in the study.

**Stimuli and apparatus.** The target images consisted of 18 pleasant and 18 unpleasant images. These images were selected from the International Affective Picture System (IAPS; [18]; see S1 Appendix) and Open Affective Standardized Image Set (OASIS; [19]; see S1 Appendix). Pleasant images contained emotionally positive objects (e.g., flowers and kittens), and the unpleasant images contained emotionally negative objects (e.g., snakes and barking dogs). All target images were resized and trimmed to 3.2˚ × 3.2˚ in MATLAB R2015a (Mathworks, Natick, MA).

Thirty-six face images were selected from Chicago face database ([20]; Asian faces; female 18 images, male 18 images; approximate age range from 20 to 40 years). All faces were converted to grayscale, resized to 3.2˚ × 3.2˚, and adjusted in mean intensity and root mean square contrast (RMS contrast) using the SHINE toolbox [21] in MATLAB R2015a (Mathworks, Natick, MA). Each face has three versions; one features a direct gaze, another has the gaze averted leftward, and the third depiction features a gaze averted rightward. The faces with the averted gaze were created by cropping and moving the pupils in the faces with the direct gaze using Photoshop (Adobe, San Jose, CA).

Stimulus presentation was controlled using a computer (running Mac OS Sierra) equipped a MATLAB and Psychtoolbox extensions. Stimuli were presented on 24-inch monitors with a resolution of 1920 × 1080 pixels at a refresh rate of 100 Hz. Responses were submitted by a keyboard ("f" and "j" keys). The keyboard was rotated 90 degrees so that the response keys were located in front or distance.

**Design.** A 3 × 2 factorial design with face type as a within-participant factor (valid-cue, invalid-cue, and no-cue) and target type as the within-participant factor (negative and positive) was conducted (Fig 1). Valid-cue faces consistently looked at the target, whereas invalid-cue faces consistently looked away from the target. No-cue faces did not move their eyes during a trial. The face images of 36 different individuals were randomly assigned to each

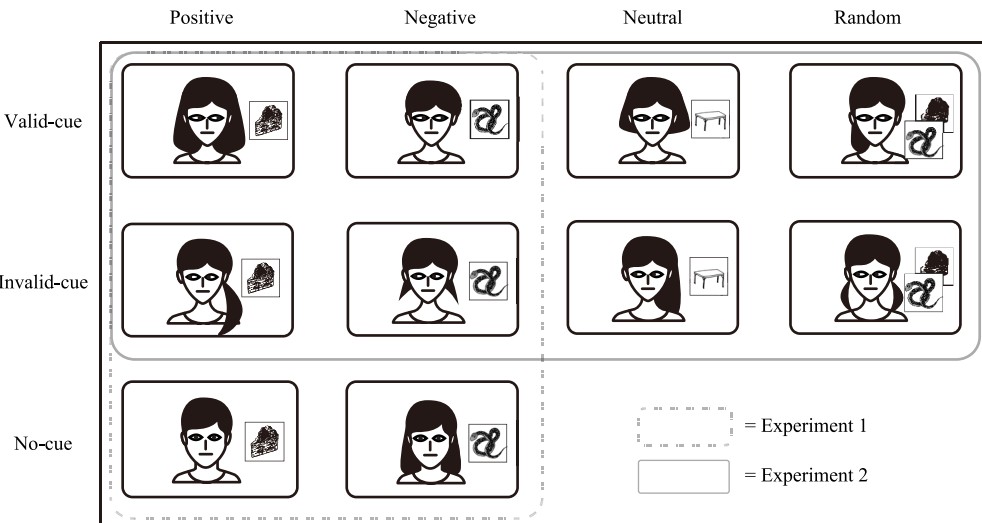

**Fig 1. Cells of Experiments 1 and 2.** The face-target pairs used in Experiment 1 are surrounded by the dotted line, and those in Experiment 2 are surrounded by the solid line. The vertical part indicates the type of face, and the horizontal part indicates the type of target. In Experiment 1, there were 6 faces (valid-cue, invalid-cue, no-cue) and targets (positive, negative). In Experiment 2, there were 8 faces (valid-cue, invalid-cue) and targets (positive, negative, neutral, random). The faces and the targets in Fig 1 are not the original images that were used in our study, but schematic images used for illustrative purposes.

experimental cell for each participant, such that some faces consistently looked at positive targets (e.g., valid-cue, positive target), whereas other faces never looked at positive targets (invalid-cue, positive target), and so on. Each experimental cell contained six faces, and the gender ratios of these faces were consistent across each experimental cell. Since the pairs of faces and targets were randomized for each trial, the face identities did not predict if a particular image would appear as the target, whereas they predicted the valence of those images.

**Procedure.** Participants were seated 60 cm from a computer display. On each trial, a fixation cross was presented for 600 ms; then a face image with a straight gaze appeared for 1,500 ms. Next, the eyes reflected a left or a right move, which was followed by a target appearing next to the face. The participant's task was to indicate the location of the target by a keypress. Mappings of target locations to keys, and of keys to response hands, were counterbalanced across participants. The face and the target remained on the screen until the participant responded or 3,000 ms had passed (see Fig 2A: a gaze-cue phase). Following 10 practice trials, participants completed six blocks of 36 trials. Each of the thirty-six faces was presented once per block. In the final block, the procedure changed (see Fig 2B: gaze-cue and evaluation phase). In this block, immediately after the participants responded to a target image, a scale from 1 to 9 appeared and, the participant had to evaluate perceived trustworthiness of the face (from 1 = *untrustworthy* to 9 = *trustworthy*). In sum, a participant completed 216 gaze-cueing trials and evaluated 36 faces.

## Results

**Gaze cueing.** Trials with incorrect responses were excluded from the analysis. The gaze-cueing trials in the gaze-cue phase were used for analysis. Accuracy was very high ($> 99.5\%$). The response times were not normally distributed. We performed a Kolmogorov-Smirnov test to check if the distribution of the response times were statistically normal. The test rejected the normality hypothesis for response times of both experiments (Experiment 1: $D = 0.15$,

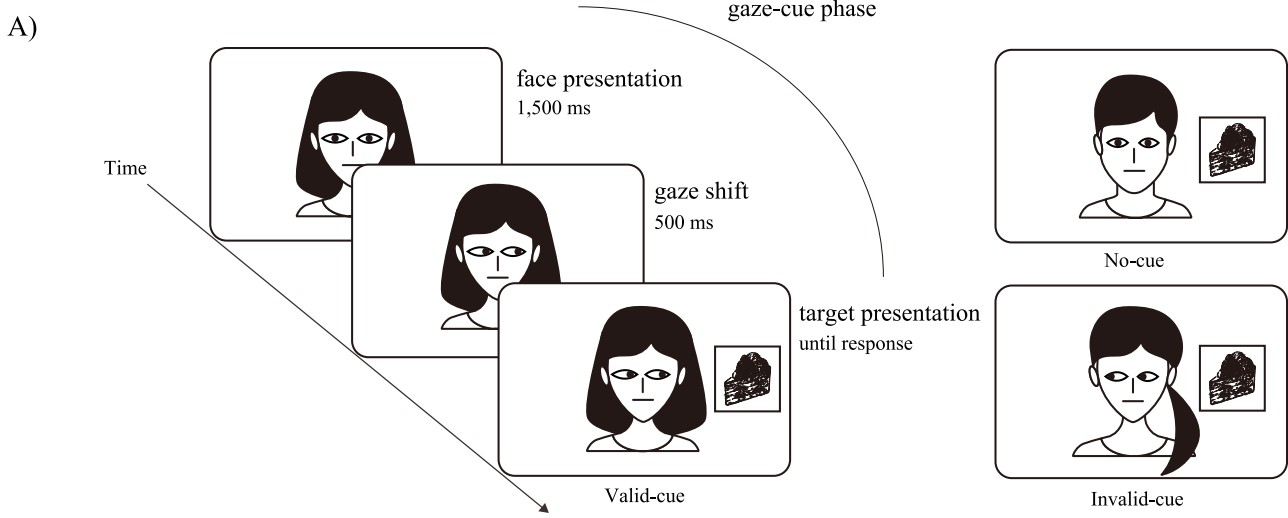

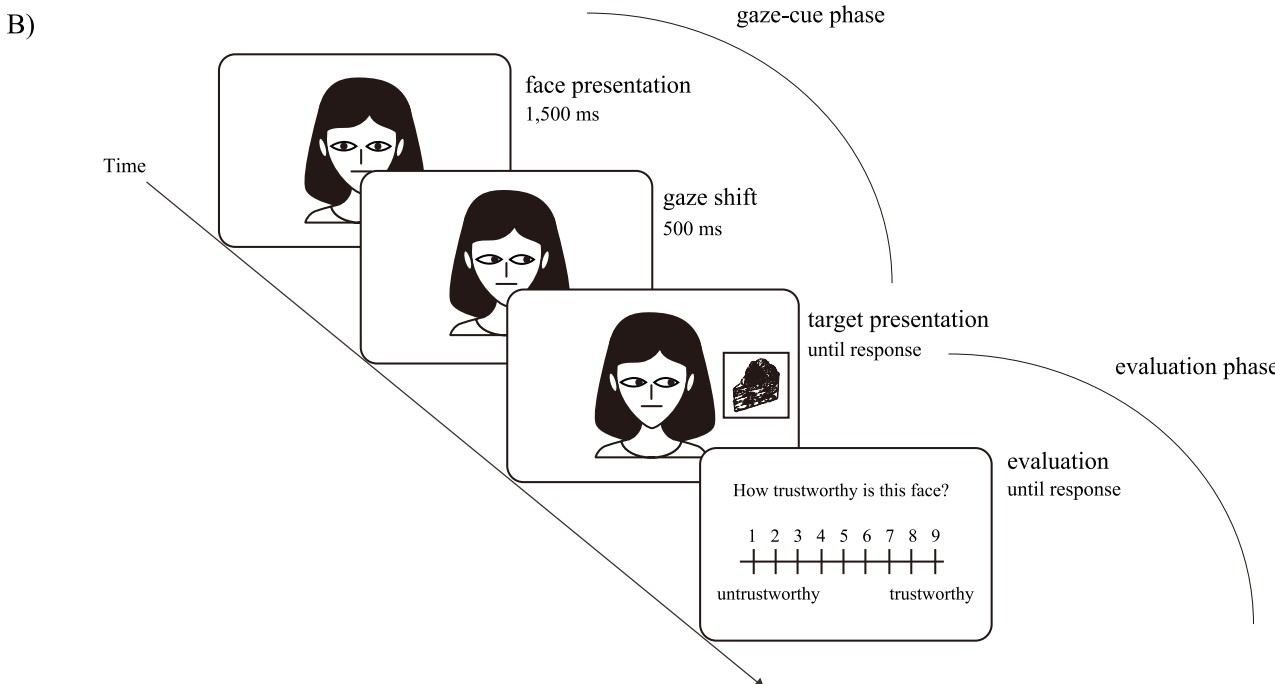

**Fig 2. Each trial of the gaze cueing task.** The faces and the targets in Fig 2 are not the original images that were used in our study, but schematic images used for illustrative purposes. (A) From one to five blocks of the gaze-cueing task only consist of the gaze-cue phase. (B) A final block consists of gaze-cue and evaluation phases.

$p < .001$; Experiment 2: $D = 0.12$, $p < .001$). Therefore, we used the median value as the summary statistic of each participant's reaction times, which decreased the sensitivity to outliers.

Fig 3 shows means of the median response times for each face type and each target type. The response times were subjected to a two-way repeated measures ANOVA with factors of face type (valid-cue, invalid-cue, and no-cue) and target type (negative and positive). The main effect of face type was significant, $F(1, 19) = 70.23$, $p < .001$, $\eta_p^2 = .79$. Multiple comparisons using the Holm method showed that judgments of participants were more rapid on valid

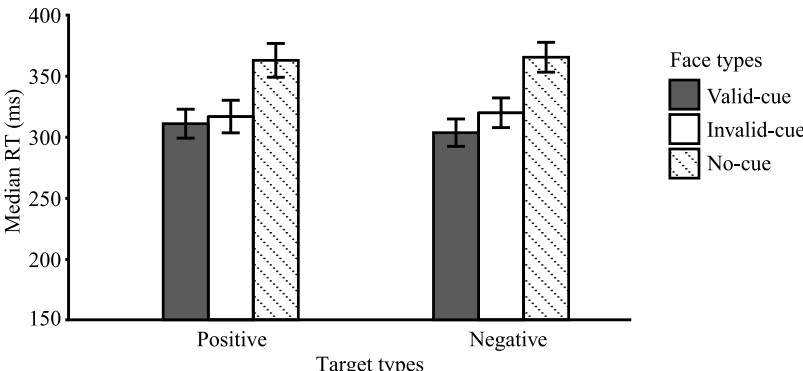

**Fig 3. Means of median response times for each type of gaze and emotional valence in Experiment 1.** Error bars represent the standard error of the mean (*SEM*).

cue trials (*M* = 307.50, *SD* = 55.90) than on invalid cue trials (*M* = 318.50, *SD* = 61.70), *t*(19) = 2.83, *p* < .05, *d* = 0.19. This indicates that the gaze cues directed attention automatically to the gazed location. Moreover, the RTs on no-cue trials (*M* = 364.20 ms, *SD* = 63.10) were slower than RTs on valid (*t*(19) = 9.43, *p* < .001, *d* = 0.95) and invalid cue trials (*t*(19) = 8.99, *p* < .001, *d* = 0.73). This delay on no-cue trials may be due to the lack of timing cue for the occurrence of the targets. Neither the main effect of target type (*F*(1, 19) = 0.04, *p* = .84, $\eta_p^2$ = .002) nor its interaction with face type were statistically significant (*F*(1, 19) = 3.06, *p* = .10, $\eta_p^2$ = .14). This suggests that the emotional valence of the target images did not affect the gaze-cueing effects.

We conducted an ANOVA for the no-cue condition described above. However, nearly all gaze cueing studies examining the effect of gaze cues have directly contrasted response times between valid- and invalid-cue conditions (e.g., [6, 8, 9]). Therefore, we additionally conducted an ANOVA with face type (valid-cue and invalid-cue) and target type (negative and positive), similar to many previous studies. Results indicated that the main effect of face type (*F*(1, 19) = 8.01, *p* = .01, $\eta_p^2$ = .30) and its interaction with target type were significant (*F*(1, 19) = 4.42, *p* = .05, $\eta_p^2$ = .19). Multiple comparisons showed that the response times were faster for valid than for invalid cue trials when the target was negative (*F*(1, 19) = 11.29, *p* < .01, $\eta_p^2$ = .37), which was not the case when the target type was positive (*F*(1, 19) = 1.75, *p* = .20, $\eta_p^2$ = .08). Moreover, the response times for valid cue trials were faster when the target type was negative than when the target type was positive (*F*(1, 19) = 6.19, *p* < .05, $\eta_p^2$ = .25). There were no significant differences in response times between positive and negative targets for invalid cue trials (*F*(1, 19) = 0.64, *p* = .43, $\eta_p^2$ = .03). Moreover, the main effect of target type was not significant, *F*(1, 19) = 0.80, *p* = .38, $\eta_p^2$ = .04.

**Facial evaluation.** Fig 4 shows trustworthiness scores of the faces for each face type and each target type. A 3 × 2 ANOVA with factors of face type (valid-cue, invalid-cue, and no-cue) and target type (positive and negative) revealed a significant main effect of face type (*F*(1, 19) = 6.58, *p* = .02, $\eta_p^2$ = .26). Post-hoc analysis showed that the invalid-cue faces (*M* = 4.10, *SD* = 1.19) were evaluated as more untrustworthy than the valid-cue (*M* = 5.51, *SD* = 1.57) and no-cue faces (*M* = 4.91, *SD* = 0.86), *t*(19) = 2.66, *p* < .05, *d* = 1.01, *t*(19) = 3.53, *p* < .01, *d* = 0.77. The difference between the valid-cue and no-cue faces did not reach significance, *t*(19) = 1.73, *p* = .10, *d* = 0.48. The main effect of target type was not significant, *F*(1, 19) = 2.26, *p* = .15, $\eta_p^2$ = .11. While numerically there appears to be a difference between the positive and negative targets, the two-way interaction between face type and target type was not significant, *F*(1, 19) = 3.41, *p* = .08, $\eta_p^2$ = .15.

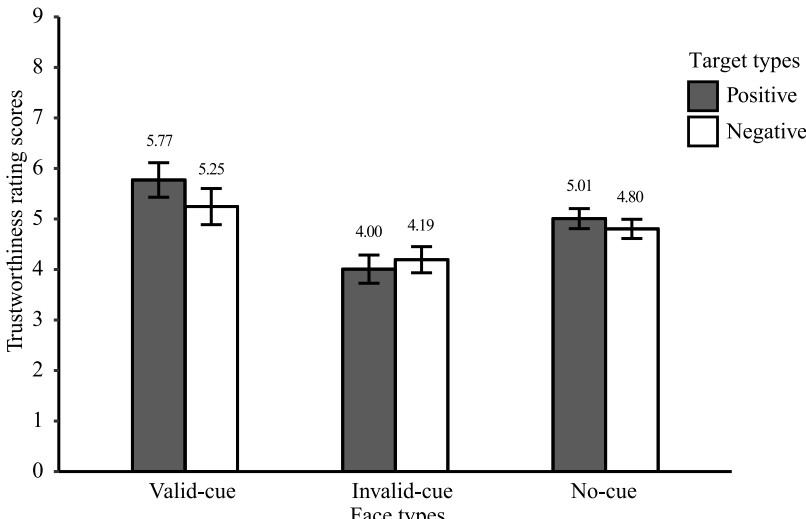

**Fig 4. Facial trustworthiness for each type of gaze and emotional valence in Experiment 1.** Error bars represent the standard error of the mean (*SEM*). Means of trustworthiness scores for each condition are shown above each bar.

In order to further explore the extent to which our data support the hypothesis that the emotional valence of the gazed images affects facial evaluation, we conducted a Bayesian paired sample *t*-test on the differences in facial evaluation among each pair of the face (valid-cue, invalid-cue, and no-cue) and target types (positive and negative) by using JASP [22]. The prior distribution was Cauchy distribution with a default scale 0.707.

Results indicated that the $BF_{10}$ for the comparison of valid faces between target types (positive and negative) was 12.86 (see Table 1), which provided strong support for the hypothesis that the emotional valence of gazed images modulated facial trustworthiness. Also, the value of $BF_{10}$ for the comparison of the positive target image between valid and invalid conditions was 9.63. This finding moderately supported the hypothesis that faces looking away from positive images are judged as more untrustworthy than faces looking at positive images. Furthermore, the comparison of the positive target between invalid and no-cue conditions indicated a $BF_{10}$ value of 67.49, which strongly supported the hypothesis that faces looking away from positive

**Table 1. Results of Bayesian paired samples t-tests in Experiment 1.**

| Faces | Targets | $BF_{01}$ | $BF_{10}$ |
|---|---|---|---|
| Invalid | Positive *vs.* Negative | 2.452 | 0.408 |
| Valid | Positive *vs.* Negative | 0.078 | 12.858 |
| No-cue | Positive *vs.* Negative | 3.261 | 0.307 |
| **Targets** | **Faces** | $BF_{01}$ | $BF_{10}$ |
| Positive | Valid *vs.* Invalid | 0.104 | 9.626 |
| | Valid *vs.* No-cue | 1.106 | 0.904 |
| | Invalid *vs.* No-cue | 0.015 | 67.489 |
| Negative | Valid *vs.* Invalid | 0.889 | 1.125 |
| | Valid *vs.* No-cue | 1.999 | 0.500 |
| | Invalid *vs.* No-cue | 0.015 | 0.003 |

[a] $BF_{01}$ indicates support for the null hypothesis.

[b] $BF_{10}$ indicates alternative hypothesis relative to the null hypothesis.

images are evaluated as more untrustworthy than no-cue faces. These findings demonstrated that the observed gaze direction modulated the effect of emotional images on facial evaluation.

## Discussion

Our results demonstrated that faces that consistently looked at positive target images were evaluated as more trustworthy than faces that consistently looked at negative images. Importantly, the emotional valence of target images affected the perceived trustworthiness of only the valid faces, and not that of invalid faces, or no-cue faces. That is, the emotional valence of target images affected the evaluation of faces only when faces looked at the target images, suggesting that gaze perception could mediate the modulation of perceived impressions regarding faces.

The lack of a trustworthiness modulation for the no-cue faces seems inconsistent with previous studies of evaluative conditioning. Prior studies have demonstrated that the perceived trustworthiness of a face was modulated by repeated, paired presentations of emotional images (e.g., [4, 5]). This inconsistency might be due to variations in gaze cue types. That is, typical experiments of evaluative conditioning have often used only faces with a direct gaze, whereas we have used faces with both an averted and a direct gaze. Under the latter circumstances, the no-cue faces might be evaluated as those that were not interested in the images because gaze shifting represents interest in objects or events [23]. This would, in turn, weaken the association between the emotional valence of the images and the faces.

## Experiment 2

In Experiment 1, we demonstrated that facial trustworthiness of faces with valid-cue was modulated by the emotional valence of target images. However, two issues were not clarified. First, because there was no baseline condition of emotional valence, it was unclear which emotional types of the targets (i.e., either or both of positive and negative valence of a target) modulated the evaluation of the valid-cue faces. Therefore, in Experiment 2, we included emotionally neutral images condition as a baseline. If the positive valence of the targets influences the trustworthiness of valid-cue faces, then the faces that always looked at the positive targets would be evaluated as more trustworthy than the faces that always looked at the neutral targets. Also, if the negative valence of the targets influences the evaluation of valid-cue faces, the faces that consistently looked at the negative targets would be evaluated as more untrustworthy than the faces that consistently looked at the neutral targets. Moreover, if both positive and negative valence of the targets influences facial evaluation, then the valid-cue faces would be evaluated as more positive in the order of positive, neutral, and negative emotional valence.

The second issue concerns the fact that the predicted emotional types by each face were fixed within the gaze-cueing tasks. Previous studies have suggested that people with consistent attitudes and behaviors tend to be evaluated as having superior personality and intelligence [24, 25]. Therefore, the consistency of gaze patterns and/or the emotional type of the target might be critical in modulating perceived trustworthiness. To address this issue, we added two types of faces; One is the faces that always looked to the target location but did not predict the same valence. The other was the faces that always looked away from the target location but did not predict the same valence. That is, these faces were predictive of the positions but not of the valences of the paired targets. By comparing the evaluation of valid-cue faces involving a condition in which the faces consistently predict a neutral valence of the targets and a condition in which the faces do not predict the valence of the targets, we would be able to clarify whether the consistency of the target valence affects the trustworthiness of faces, independently of the emotional valence for the target. Hence, we predicted that if the consistency of the target

valence influences facial trustworthiness, the valid-cue faces that predicted neutral valence type of targets would be evaluated as more trustworthy than the valid-cue faces that did not predict (i.e., randomly looked) the valence types of the targets. On the other hand, if the effect of the consistency of the target valence is not included in the effect of gaze-induced trustworthiness in Experiment 1, then there should be no differences in the evaluation for the valid-cue faces between the condition in which the faces predict the neutral valence of the targets and the condition in which the faces do not predict the emotional valence of the targets.

## Materials and methods

**Participants.**   We conducted a power analysis (G*power; [16, 17]), which indicated that a minimum of 14 participants was necessary to achieve a power level of 0.80 based on the same criteria as the power calculation of Experiment 1 (except for 8 measurements and a nonsphericity correction of 0.14). Finally, twenty-four undergraduate students (2 males and 22 females, mean age = 20.75 years) participated in Experiment 2. All participants reported having normal vision or corrected-to-normal vision and provided informed consent.

**Stimuli and apparatus.**   Eighteen neutral images were newly selected from OASIS [19], which consisted of neutral scenes, such as a cityscape or a natural scene. They were resized and trimmed to 3.2˚ × 3.2˚ in MATLAB R2015a (Mathworks, Natick, MA). The pleasant and unpleasant images were identical to those of Experiment 1. Thus, the target images consisted of 18 positive, 18 negative, and 18 neutral images.

Twelve new faces were taken from Chicago face database [20] and added to the faces of Experiment 1. Thus, forty-eight Asian faces (female 24 images, male 24 images; approximate age range from 20 to 40 years) were used in Experiment 2.

The experimental apparatus for the stimulus presentation was identical to that of Experiment 1.

**Design.**   For each participant, the faces were randomly assigned to one of the eight conditions of a 2 (gaze-cue: valid, invalid) × 4 (valence-target: positive, negative, neutral, random) factorial design (Fig 1). For example, those faces with a valid position-cue always directed a participant's gaze to the target position, whereas those with an invalid position-cue consistently directed the gaze away from the target. Moreover, faces in the neutral valence-target condition were always followed by the targets of emotionally neutral images. By contrast, the faces in the random valence-target condition were followed by either positive or negative images with equal probability.

**Procedure.**   The procedure was identical to that of Experiment 1, with the exception that the participants completed the gaze-cue phase comprising six blocks of 48 trials each, after the 10 practice trials. Thus, they completed 384 gaze-cueing trials. Following the gaze-cue phase, the participants engaged in the evaluation phase that comprised two blocks of 48 trials each. In this evaluation phase, all faces were rated two times. The evaluation scores for these two times were averaged; this, then became the trustworthiness score of each face. Furthermore, the evaluation scores for the faces in the random condition were calculated separately when the positive target was presented in the evaluation phase and when the negative target was presented.

## Results

**Gaze cueing.**   The analysis indicated that accuracy in the gaze-cueing trials was very high (> 98.2%). Incorrect responses were removed from further analysis.

Fig 5 shows the median response times for each prediction type in the valid and invalid conditions. Response times were subjected to a two-way ANOVA with factors of type of gaze (valid or invalid) and target type (positive, negative, neutral, random). The main effect

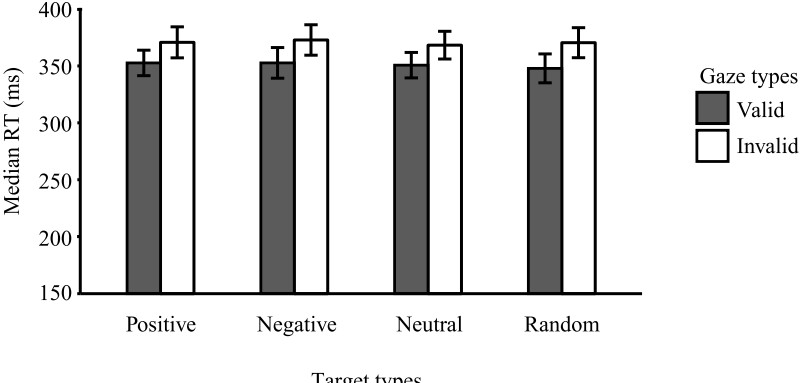

**Fig 5. Median response times for each type of gaze and target in Experiment 2.** Error bars represent the standard error of the mean (*SEM*).

involving the type of face gaze was significant, $F(1, 23) = 21.14$, $p < .001$, $\eta_p^2 = .48$, indicating that the RTs of the valid cue trials were faster than those of the invalid cue trials. Neither the main effect of target type, $F(1, 23) = 0.85$, $p = .37$, $\eta_p^2 = .04$, nor the interaction with gaze validity was significant, $F(1, 23) = 0.29$, $p = .60$, $\eta_p^2 = .01$. These findings indicate that the target valence did not affect the gaze cueing effect.

**Facial evaluation.** Fig 6 shows the trustworthiness rating scores for each position-cue and valence-target type. The trustworthiness scores were subjected to a two-way ANOVA with factors of position-cue (valid and invalid) × valence-target (positive, negative, neutral, random), which revealed a significant main effect of position-cue, $F(1, 23) = 13.03$, $p < .01$, $\eta_p^2 = .36$, suggesting that the valid position-cue faces were evaluated as more trustworthy than the invalid position-cue faces. Moreover, a main effect of valence-target was significant, $F(1, 23) = 4.77$, $p < .05$, $\eta_p^2 = .17$, but the post-hoc analysis showed no significant differences among cue types. The interactions between valence-target and position cues were not significant,

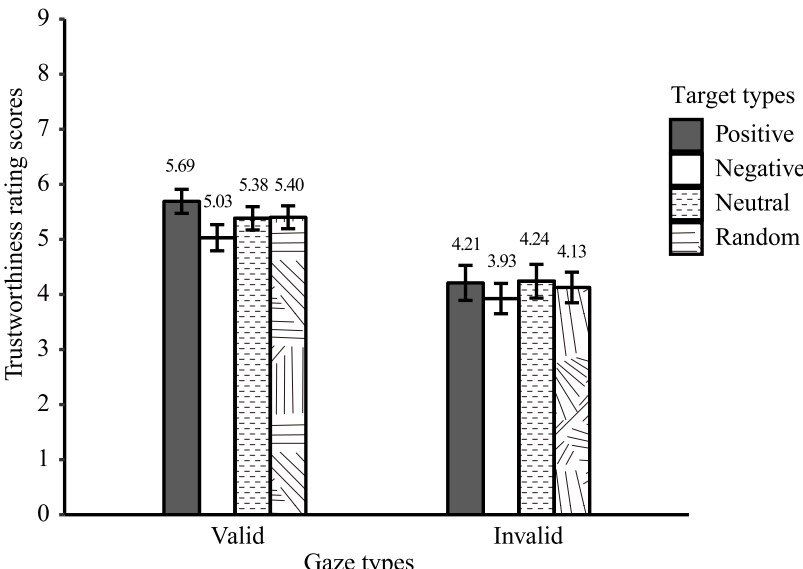

**Fig 6. Trustworthiness for each type of gazes and targets in Experiment 2.** Error bars represent the standard error of the mean (*SEM*). Means of trustworthiness scores for each condition are shown above each bar.

**Table 2. Results of Bayesian paired sample *t*-test for valid faces.**

| Position-cue | Valence-target | $BF_{01}$ | $BF_{10}$ |
|---|---|---|---|
| Valid | Positive *vs*. Negative | 0.15 | 6.82 |
| | Positive *vs*. Neutral | 0.38 | 2.62 |
| | Positive *vs*. Random | 0.92 | 1.09 |
| | Negative *vs*. Neutral | 0.53 | 1.88 |
| | Negative *vs*. Random | 0.39 | 2.54 |
| | Neutral *vs*. Random | 4.60 | 0.22 |
| Invalid | Positive *vs*. Negative | 1.39 | 0.72 |
| | Positive *vs*. Neutral | 4.54 | 0.22 |
| | Positive *vs*. Random | 3.68 | 0.27 |
| | Negative *vs*. Neutral | 1.45 | 0.69 |
| | Negative *vs*. Random | 1.13 | 0.88 |
| | Neutral *vs*. Random | 3.31 | 0.30 |

$F(1, 23) = 2.36$, $p = .14$, $\eta_p^2 = .09$. These results suggest that the target position cues modulated the perceived trustworthiness of a face.

It is possible that the number of evaluations, i.e., twice per face in Experiment 2, weakened the result by repeating the evaluation and averaging the two facial evaluations scores. To exclude this possibility, we analyzed the differences in facial evaluations between the first and the second evaluation by using Welch's *t*-test. The results showed that differences between the number of evaluations were neither significant in cases in which faces gazed toward negative, $t(45.27) = -0.49$, $p = .62$, nor positive images, $t(45.94) = 0.11$, $p = .92$, suggesting that the number of evaluations did not affect facial evaluations in Experiment 2.

Table 2 shows the Bayesian analysis results for differences in the trustworthiness scores of faces among position-cue (valid and invalid) and valence-target types (positive, negative, neutral, random). We performed the Bayesian analysis by using JASP [22]. The setting of the prior distribution was identical to Experiment 1.

Results indicated that $BF_{10}$ for comparing positive and negative valence-target conditions for valid faces was 6.82, which moderately supported the alternative hypothesis. This result supported our contention that faces always looking at positive images are evaluated as more trustworthy than faces always looking at negative images. The $BF_{10}$ for the comparison of valid faces between positive and neutral valence-targets and between negative and neutral valence-targets was smaller than 3, which indicates weak evidence. Therefore, we could not firmly conclude that there was an effect of the emotional valence of each target on the facial evaluation.

Importantly, we also compared facial evaluations between neutral and random valence-target conditions. Results revealed that $BF_{01}$ for the valid faces between the neutral and random valence-target conditions was 4.60, which provided moderate support for the null hypothesis. Thus, we found that the consistency of the target valence itself did not affect trustworthiness evaluations of the faces.

**Additional analysis: Can increased facial trustworthiness be explained as a response to a single or an iterative presentation of a face and an image?.** One may argue that the trustworthiness of the valid faces was affected only by a single presentation of the face and the image that preceded the rating response in the evaluation phase rather than by repetitive presentation of the face and the image during the gaze-cue phase. To clarify whether this is the case, we compared the evaluation scores of the valid-random faces with those of the valid-positive and valid-negative faces. Because the valid faces in the random conditions looked at the positive or the negative target with equal probability, we can assume that the perceived

trustworthiness of the valid-random faces was not strongly modulated by repetitive presentations in the gaze-cue phase compared to those of the valid-positive and valid-negative faces. Therefore, if the positive valence of the gazed images is repeatedly encoded in the face images to affect facial evaluation, then the faces that always looked at the positive targets would be rated as more trustworthy than the random faces that looked at the positive target in the evaluation phase.

The $BF_{10}$ supporting the hypothesis that valid faces in the positive condition were rated as more trustworthy than valid faces in the random condition looking at positive targets was 1.44. Furthermore, $BF_{10}$ supporting the hypothesis that valid faces in the negative condition were evaluated as more untrustworthy than valid faces in the random condition looking at negative targets was 1.77. These results, if anything, would prefer the interpretation that the modulation of the perceived trustworthiness of the face was due to the learned association between the face and target, not due to a single presentation that preceded the rating response. Of course, since these interpretations were based on the weak evidence, we cannot strongly conclude how to increase the facial evaluation.

## Discussion

The findings of Experiment 2, similar to those of Experiment 1, demonstrated that faces consistently looking at positive targets were evaluated as more trustworthy than faces consistently looking at negative targets. Moreover, valid-cue faces with neutral targets were regarded as more trustworthy than valid-cue faces with negative targets, which were less trustworthy than valid-cue faces with positive targets. The Bayesian analyses supported these tendencies, suggesting that both emotional valences impacted facial trustworthiness. Also, the perceived trustworthiness in the neutral valence-target condition was not higher than that in the random valence-target condition. These results suggest that the consistency of the target valence did not modulate facial evaluation. This might be because the gaze direction, either looking towards or away from a target was always consistent within each face identity. The consistency of gaze direction might be more dominant in modulating perceived facial trustworthiness than the consistency of target valence. Therefore, consistent gaze direction might be prioritized in determining facial trustworthiness, which weakens the effect of target valence consistency. It is suggested that future studies examine this possibility.

## General discussion

The present study was designed to clarify whether the emotional valence of a gazed target affected the gazer's trustworthiness. Experiment 1 indicated faces that consistently looked toward positive images were evaluated as more trustworthy than faces that consistently looked toward negative images. A similar pattern was obtained in Experiment 2. Furthermore, we found that the consistency of the valence of a target was unrelated to the improved evaluation of the gazer, suggesting that the emotional valence of target images influences evaluations of the gazer through the perception of the gaze direction. Importantly, faces that consistently looked away from positive images were rated as less trustworthy than faces that consistently looked at positive images or faces in the no-cue condition. In general, looking away from positive objects/events is not adaptive. It is plausible that such incomprehensible behavior might lead to the evaluation of low trustworthiness for the person. However, we cannot disregard the possibility that the emotional information in the images boosted trustworthiness or enhanced another positive affect such as attractiveness. In future research, we should examine the types of impressions that are influenced by the emotional valence of gazed images or events.

This study demonstrated that the emotional valence of gazed at images affected facial evaluation through gaze cues. The relationship between faces and gazed images, which were learned during the gaze-cue phases, changed facial evaluations. Consistent with this possibility, Kirkham et al. suggested that participants learn to trust or distrust people based on what they look at and how they respond [26]. They demonstrated that faces smiling for a positive scene were evaluated as more trustworthy than faces frowning for a positive scene, suggesting that encoding the face-scene consistency influenced perceived facial trustworthiness. In the light of previous work, our findings suggest that consistent behavioral signals, such as the gaze direction, enhance perceived facial trustworthiness even in the absence of emotional expressions.

An exciting aspect of our results rests in the finding that the effect of positive target images on valid-cue faces was higher than the effect of negative targets on valid-cue faces. This result suggests that facial evaluation based on the gaze cue was primarily modulated by positive valence. Such prioritization of positive valence in joint attention has been suggested by different studies [9, 27, 28]. For example, facial trustworthiness caused by gaze perception was better when a face type was happy than when it was neutral or angry [9]. Moreover, the gazed objects were evaluated as more likable than ignored objects, but only if the faces were smiling [27]. These findings support the idea that gaze monitoring is a fundamental process that plays a critical role in positive social situations such as collaboration and friendship [28, 29]. Our results provide further empirical evidence supporting this view.

Relative to Experiment 1, the facial trustworthiness was not strongly influenced by the emotional valence of the gazed images in Experiment 2. There are at least two possible explanations for these findings. The first is related to the fact that the number of face images in Experiment 2 (48 images) was more substantial than that in Experiment 1 (36 images). As a result, participants may have been unable to memorize the pairing between faces and emotional information. A second possibility concerns the consistency of the valence of the target images. Experiment 2 included faces that did not predict the target valence. Such faces with low predictability of the target valence may weaken the association between the faces and emotional valence, with the result that emotional information did not strongly affect the trustworthiness of any of the faces.

In summary, this study demonstrates that emotional information associated with gazed images influences the facial trustworthiness of the gazer. When judging whether another individual is cooperative or not, we need to carefully scrutinize whether the outcome of the other's actions is beneficial. It is plausible that we encode the faces that gaze at an object together with information about the gazed objects. In conclusion, this study demonstrated that gaze patterns and the emotional valence of the objects are used for identifying the trustworthiness of faces.

## Supporting information

**S1 Appendix. IAPS and OASIS numbers of used images.**
(DOCX)

**S2 Appendix. Data in our experiments.**
(XLSX)

## Acknowledgments

We are grateful to the anonymous reviewers for their comments on earlier version of this paper.

## Author Contributions

**Conceptualization:** Risako Shirai, Hirokazu Ogawa.

**Data curation:** Risako Shirai.

**Formal analysis:** Risako Shirai.

**Funding acquisition:** Risako Shirai.

**Investigation:** Risako Shirai.

**Methodology:** Risako Shirai.

**Project administration:** Hirokazu Ogawa.

**Resources:** Hirokazu Ogawa.

**Software:** Risako Shirai.

**Supervision:** Hirokazu Ogawa.

**Visualization:** Risako Shirai.

**Writing – original draft:** Risako Shirai, Hirokazu Ogawa.

**Writing – review & editing:** Risako Shirai, Hirokazu Ogawa.

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
