## [Decision Letter · Decision Letter 0]

23 Jul 2019

PONE-D-19-14867

Affective evaluation of images influences personality judgments through gaze perception

PLOS ONE

Dear Dr. Shirai,

Thank you for submitting your manuscript to PLOS ONE. After careful consideration, we feel that it has merit but does not fully meet PLOS ONE’s publication criteria as it currently stands. Therefore, we invite you to submit a revised version of the manuscript that addresses the points raised during the review process.

As you will see below, the reviewers provided insightful feedback on your manuscript. However, they also raised a number of important concerns, especially with regard to methodological, statistical, and interpretational aspects of your manuscript. These include insufficient details on the stimuli and apparatus; issues with the power calculation and sample size; and issues with how marginally significant effects are interpreted. I urge you to pay close attention to their comments in your revision.

We would appreciate receiving your revised manuscript by September 21, 2019. To enhance the reproducibility of your results, we recommend that if applicable you deposit your laboratory protocols in protocols.io, where a protocol can be assigned its own identifier (DOI) such that it can be cited independently in the future. For instructions see: http://journals.plos.org/plosone/s/submission-guidelines#loc-laboratory-protocols

We look forward to receiving your revised manuscript.

Kind regards,

Veronica Whitford, Ph.D.

Academic Editor

PLOS ONE

Journal Requirements:

2) In your data availability statement you write, "All relevant data are within the paper and its Supporting Information files." Please ensure you have provided the individual data points used to create the figures and determine means, medians and variance measures presented in the results, tables and figures (http://journals.plos.org/plosone/s/data-availability#loc-faqs-for-data-policy). If these data cannot be publicly deposited or included in the supporting information, e.g. due to patient privacy or ownership by a third party, explain why and explain how researchers may access them.

3) We note that Figure 1 includes an image of a patient / participant in the study. 

Please respond by return e-mail with an amended manuscript. We can upload this to your submission on your behalf.

If you are unable to obtain consent from the subject of the photograph, please either instruct us to remove the figure or supply a replacement figure by return e-mail for which you hold the relevant copyright permissions and subject consents. In some cases, you may need to specify in the text that the image used in the figure is not the original image used in the study, but a similar image used for illustrative purposes only. We can make any changes on your behalf.

4) Please include your tables as part of your main manuscript and remove the individual files. Please note that supplementary tables (should remain/ be uploaded) as separate "supporting information" files

Reviewers' comments:

Reviewer's Responses to Questions

**Comments to the Author**

1. Is the manuscript technically sound, and do the data support the conclusions?

Reviewer #1: Partly

Reviewer #2: No

2. Has the statistical analysis been performed appropriately and rigorously? 

Reviewer #1: Yes

Reviewer #2: No

3. Have the authors made all data underlying the findings in their manuscript fully available?

Reviewer #1: No

Reviewer #2: Yes

4. Is the manuscript presented in an intelligible fashion and written in standard English?

Reviewer #1: Yes

Reviewer #2: Yes

5. Review Comments to the Author

Reviewer #1: 1) The authors should explain why they chose to use Cohen’s f for their sample size calculation, and why they chose a medium effect size. Were these choices based on previous published data? I tried to replicate the sample size calculation the authors employed in E1 using G*Power, however they did not provide all the information needed to run the calculation. If I assume the statistical test is a repeated measures ANOVA, within factors, and select 1 group and 6 measurements, my result it close to the one the authors obtained (19 participants instead of 18), however it would be good if the authors could clarify their choices here. In E2, could the authors explain the rationale for why they chose a sample size much larger than the sample size returned by their effect size calculation.

2) For the sake of completeness, the authors should provide the image numbers for all 36 images used in the study in the Appendix, instead of just a subset of the images.

3) For the faces, how were the eyes altered to depict leftward and rightward gaze? Were the faces matched for attractiveness? Previous research shows that more attractive individuals are seen in a more positive light (e.g., Bascandziev & Harris, 2014), thus it is imperative to control for attractiveness in these types of studies.

4) The Methods section is currently lacking information on the Apparatus used to present the computer task, along with the Design section to indicate the conditions, etc. Please provide details as to the computer, monitor, program used to present the task, etc.

5) Which keys were used to respond to the target? This should be noted in the manuscript, due to the well-known Simon effect. This effect, whereby a left key press made by a left hand to signal a left location, leads to a congruent spatial stimulus-response mapping situation, which makes it difficult to tease apart whether it is attention or motor preparation that is leading to the RT effects. As there is some evidence that social attention is affected during Simon tasks (e.g., McKee, Christie, & Klein, 2007; Zorzi et al., 2003), the authors should be mindful of the conclusions they draw depending on how the responses were provided.

6) Can the authors explain their choice to use median RTs instead of the more conventional mean RTs?

7) Figures 2 and 5. The results would be clearer if the y-axis was scaled to start around 300ms instead of zero, and go until around 400ms instead of 600ms.

8) page 11, line 176. How is the finding (two-way interaction) deemed marginal with a p value of 0.08, whereas on the next page, lines 188 and 189 the effects are deemed to be “no effect” with p values of 0.09 and 0.08. The authors should avoid usage of “marginal”; if using NHST, then something is either significant or not.

9) line 223 – the word “thus” is repeated twice.

10) page 16, lines 253-255: “Second, because the valid-cue faces were predictive of both the location and valence of the targets, we could not determine which type of predictability was critical for the perceived trustworthiness to be modulated.”

To clarify, aren’t both the valid-cue and invalid-cue faces predictive of location and valence? Specifically, the invalid-cue faces will ALWAYS look away from the target location (which needs to be implicitly learned just as in the valid-cue face condition), and certain faces predict positive images while others predict negative images, no? Thus, the difference between valid-cue and invalid-cue faces is the ‘helpfulness’ participants attribute to the gaze information (and thus the face itself). I suggest revising this sentence to incorporate this more nuanced difference.

11) I appreciated that the authors collected two measures of trustworthiness for each face in E2. I also wonder, however, if the single measure of trustworthiness done in E1 could be driving the difference between ratings for positive and negative images in the valid-cue condition. One way the authors could partially mitigate this concern would be to calculate the standard deviation across the two scores per face in E2, and demonstrate that the variability is low. This would give additional weight to the conclusions drawn from E1.

12) Figures 3 and 6. For reader ease, the authors should add the average trustworthiness rating scores for each condition above the corresponding bar.

13) I cannot find the repository where all of the raw data are available. Can the authors please provide this information.

Reviewer #2: This paper aimed at investigating whether the gaze cueing effect and the personality judgment of a face would be affected by the valence of a target. To this aim, participants were presented with a standard gaze cuing procedure in which the affective valence of the response target was manipulated. After the gaze cuing procedure, participants were asked to judge how trustworthy they perceived the face stimuli. The results, on which I comment more below, were interpreted by the Authors as to show that valid faces that looked at the positive target were evaluated as more trustworthy than the faces that looked at the negative targets.

GENERAL COMMENTS

The research question is introduced clearly and addresses an intriguing issue. Unfortunately, several limitations prevent me from supporting the publication of this study.

My main concerns relate to the sample and the statistical approach. The rationale beyond the power analysis to establish the sample sizes of both experiments (i.e., 20 and 24 participants, respectively) is unclear. For example, it is unclear how the reference effect size of 0.25 was established (e.g., previous research?) and whether the Authors used an a-priori or post-hoc analysis. Additionally, given the subtle modulation that this study aims at investigating, the resulting sample sizes appear small – especially considering the relatively small number of trials that were used (i.e., 216).

Possibly as a result of this relative lack of power, the results are uncertain and sometimes not statistically significant for the central effects of interest. The additional Bayesian analyses are, in my opinion, insufficient to make the results more reliable – which seems the only or main reason why they were added. I agree that Bayesian analyses can be highly informative, and sometimes even preferable to the NHST approach. However, their rationale should be explained clearly, and more details should be given on how they were implemented (e.g., description of priors).

In terms of overall results, the only reliable finding I can appreciate is that validly cuing faces were perceived as more trustworthy than invalidly cuing faces. This finding is consistent with previous literature but is not informative to Authors’ hypotheses.

Thus, overall, I think that this study is not ready for publication. However, I also think that this paper pursues an interesting line of research, and I wish the Authors the best of luck with the investigation on this topic. Below are some additional comments (in no particular order) that I hope will benefit Authors’ future work.

DETAILED COMMENTS

The expression “marginally significant” is inappropriate (page 11, line 176) as it does not reflect the arbitrary nature of the p-value. Furthermore, it appears that Authors use this label arbitrarily, as for example similar results (p=.08, eta=.15) are described as “marginally significant” (page 11, line 176) for the two-way interaction of interest and considered non-significant a few lines below (page 12, line 189). Thus, the section exploring the two-way interaction should be eliminated.

Related to the first point, the inclusion of correlations does not seem appropriate (at least with the separation by target type) because they follow results that appear unreliable (i.e., the interaction). Additionally, the correlations are not significant; thus, it is unclear what the value of Figure 4 is.

The description of the methods of Experiment 1 is unclear, particularly on whether face validity and target valence target were manipulated independently. On first reading, I assumed that they were manipulated independently, which would be recommended. However, I reconsidered this assumption later on, when Experiment 2 was introduced. The Authors write “Second, because the valid-cue faces were predictive of both the location and valence of the targets, we could not determine which type of predictability was critical for the perceived trustworthiness to be modulated” (page 16, lines 253-255). This sentence suggests that valid-cue face always cued positive images, and this is a major issue casting doubts on the overall procedure of Experiment 1.

Experiment 2 substantially replicates the main effect of Experiment 1 and does not add much in its present form. Furthermore, the additional analyses (“Does history of target valence effect facial evaluation?” page 23, line 382) are poorly introduced/explained, and it is unclear what their contribution is, given the lack of clear and reliable results.

6. PLOS authors have the option to publish the peer review history of their article (what does this mean?). If published, this will include your full peer review and any attached files.

Reviewer #1: No

Reviewer #2: No

---

## [Author Response · Author response to Decision Letter 0]

12 Sep 2019

RESPONSE TO REVIEWER 1:

Comment 1: 

The authors should explain why they chose to use Cohen’s f for their sample size calculation, and why they chose a medium effect size. Were these choices based on previous published data?

I tried to replicate the sample size calculation the authors employed in E1 using G*Power, however they did not provide all the information needed to run the calculation. If I assume the statistical test is a repeated measures ANOVA, within factors, and select 1 group and 6 measurements, my result it close to the one the authors obtained (19 participants instead of 18), however it would be good if the authors could clarify their choices here. In E2, could the authors explain the rationale for why they chose a sample size much larger than the sample size returned by their effect size calculation.

Response 1: We thank the reviewer for informing us about the lack of information needed to run this calculation. In accordance with this comment, we have added the information about the calculation for the sample size to Materials and methods.

Obviously, we could not know the appropriate effect size for our experiments in advance, so we used the standard setting (i.e., Cohen’s f of 0.25 in G*power). Finally, we investigated a relatively large number of participants than the calculated sample size after considering the possibility of missing data. 

Comment 2: 

For the sake of completeness, the authors should provide the image numbers for all 36 images used in the study in the Appendix, instead of just a subset of the images.

Response 2: We appreciate the reviewer's comment on this point. 

We agree with the reviewer’s comment. Accordingly, we have included the image numbers for all images in the Supporting information.

Comment 3: 

For the faces, how were the eyes altered to depict leftward and rightward gaze? Were the faces matched for attractiveness? Previous research shows that more attractive individuals are seen in a more positive light (e.g., Bascandziev & Harris, 2014), thus it is imperative to control for attractiveness in these types of studies.

Response 3: We thank the reviewer for this comment. We created the faces with the averted gaze by cropping and changing the pupils of the faces with a directed gaze. We have included an explanation to the Stimulus and apparatus section providing more information about image processing. 

As this reviewer observes, we also think that the control of attractiveness of the faces is very important in this study. In our study, the face images were randomly assigned to the conditions per participants. Therefore, we believe that the effect of facial attractiveness did not significantly influence our findings.

Comment 4: 

The Methods section is currently lacking information on the Apparatus used to present the computer task, along with the Design section to indicate the conditions, etc. Please provide details as to the computer, monitor, program used to present the task, etc.

Response 4: Thank you for this comment. We agree with the reviewer’s comment. Accordingly, we have added information about the apparatus to Materials and methods.

Comment 5: 

Which keys were used to respond to the target? This should be noted in the manuscript, due to the well-known Simon effect. This effect, whereby a left key press made by a left hand to signal a left location, leads to a congruent spatial stimulus-response mapping situation, which makes it difficult to tease apart whether it is attention or motor preparation that is leading to the RT effects. As there is some evidence that social attention is affected during Simon tasks (e.g., McKee, Christie, & Klein, 2007; Zorzi et al., 2003), the authors should be mindful of the conclusions they draw depending on how the responses were provided.

Response 5: We welcome this insightful comment. We should have explained the methods of our study more clearly. Therefore, we have added an explanation about response mapping to the Materials and methods. 

Comment 6:

Can the authors explain their choice to use median RTs instead of the more conventional mean RTs?

Response 6: We wish to thank the reviewer for these comments. When we examined the distribution of the data for each participant, we found bimodality and asymmetry on the distribution of response times for each participant. Therefore, we used the median value for analysis. Several previous studies also have used median values in the analysis for the gaze cueing tasks (e.g., Bayliss, Paul, Cannon, & Tipper, 2006).

Bayliss, A. P., Paul, M. A., Cannon, P. R., & Tipper, S. P. (2006). Gaze cuing and affective judgments of objects: I like what you look at. Psychonomic Bulletin & Review, 13(6), 1061–1066.

Comment 7:

Figures 2 and 5. The results would be clearer if the y-axis was scaled to start around 300ms instead of zero, and go until around 400ms instead of 600ms.

Response 7: We thank your comments. 

To assist these interpretations for readers, we have changed the y-axis in Figure 2 and 4 (Figure 4 is new version of Figure 5).

Comment 8: 

page 11, line 176. How is the finding (two-way interaction) deemed marginal with a p value of 0.08, whereas on the next page, lines 188 and 189 the effects are deemed to be “no effect” with p values of 0.09 and 0.08. The authors should avoid usage of “marginal”; if using NHST, then something is either significant or not.

Response 8: Your reading of our interpretations is quite accurate; we appreciate your comment on this point. We also think that we should have a consistent interpretation for the p values of 0.08. Because we considered that these points should be changed, we have rewritten the interpretation for the analysis in the Results.

Comment 9: 

line 223 – the word “thus” is repeated twice.

Response 9: We thank your comments on this point. We have changed from “thus” to “this”.

Comment 10:

page 16, lines 253-255: “Second, because the valid-cue faces were predictive of both the location and valence of the targets, we could not determine which type of predictability was critical for the perceived trustworthiness to be modulated.”

To clarify, aren’t both the valid-cue and invalid-cue faces predictive of location and valence? Specifically, the invalid-cue faces will ALWAYS look away from the target location (which needs to be implicitly learned just as in the valid-cue face condition), and certain faces predict positive images while others predict negative images, no? Thus, the difference between valid-cue and invalid-cue faces is the ‘helpfulness’ participants attribute to the gaze information (and thus the face itself). I suggest revising this sentence to incorporate this more nuanced difference.

Response 10: We thank you for this insightful comment. Indeed, our writing was unclear. To help the interpretation of readers about the differences across the conditions, we have rewritten and added the explanation about these conditions to the introduction in Experiment 2.

Comment 11: I appreciated that the authors collected two measures of trustworthiness for each face in E2. I also wonder, however, if the single measure of trustworthiness done in E1 could be driving the difference between ratings for positive and negative images in the valid-cue condition. One way the authors could partially mitigate this concern would be to calculate the standard deviation across the two scores per face in E2, and demonstrate that the variability is low. This would give additional weight to the conclusions drawn from E1. 

Response 11: 

We appreciate your comment and advice. In accordance with your comments, we assessed whether there were the differences between two scores per face in Experiment 2. As a result, we did not find the differences between these scores. Therefore, we consider that the facial evaluation in Experiment 2 was not strongly influenced by the variability between the scores.

Comment 12:

Figures 3 and 6. For reader ease, the authors should add the average trustworthiness rating scores for each condition above the corresponding bar.

Response 12:

Thank you for your comment. In accordance with your idea, we have added the rating scores to Figures 3 and 5 (Figure 5 is new version of Figure 6).

Comment 13: 

I cannot find the repository where all of the raw data are available. Can the authors please provide this information.

Response 13:

We thank you for your comment. We have added the requested information to the Supporting information.

 

RESPONSE TO REVIEWER 2:

Comment 1: 

The research question is introduced clearly and addresses an intriguing issue. Unfortunately, several limitations prevent me from supporting the publication of this study.

My main concerns relate to the sample and the statistical approach. The rationale beyond the power analysis to establish the sample sizes of both experiments (i.e., 20 and 24 participants, respectively) is unclear. For example, it is unclear how the reference effect size of 0.25 was established (e.g., previous research?) and whether the Authors used an a-priori or post-hoc analysis. 

Response 1: 

We wish to express our deep appreciation to this reviewer for this insightful comment surrounding this topic. . As you wrote, the information about the sample analysis was not fully in our paper.

 We conducted a-priori power estimation in our study. Moreover, because we obviously could not know the appropriate effect size for our experiments in advance, we used the standard setting (i.e., effect size (f) of 0.25 in G*power). In accordance with your comments, we have added the relevant information about sample analysis to the Materials and methods. 

Comment 2: 

Additionally, given the subtle modulation that this study aims at investigating, the resulting sample sizes appear small – especially considering the relatively small number of trials that were used (i.e., 216).

Possibly as a result of this relative lack of power, the results are uncertain and sometimes not statistically significant for the central effects of interest. The additional Bayesian analyses are, in my opinion, insufficient to make the results more reliable – which seems the only or main reason why they were added. I agree that Bayesian analyses can be highly informative, and sometimes even preferable to the NHST approach. However, their rationale should be explained clearly, and more details should be given on how they were implemented (e.g., description of priors).

In terms of overall results, the only reliable finding I can appreciate is that validly cuing faces were perceived as more trustworthy than invalidly cuing faces. This finding is consistent with previous literature but is not informative to Authors’ hypotheses.

Thus, overall, I think that this study is not ready for publication. However, I also think that this paper pursues an interesting line of research, and I wish the Authors the best of luck with the investigation on this topic. Below are some additional comments (in no particular order) that I hope will benefit Authors’ future work.

Response 2: Thank you for your insightful comments. As you wrote, we felt that we needed to ascertain if we had enough power, taking into account the number of trials (i.e., 216 trials). Therefore, we performed the power analysis by PANGEA that is web-based power application that can estimate power using information of the trials. As a result, we obtained a power of 0.911 (PANGEA default effect size [d]: 0.45, iterations: 36 times, participants: 20). We consider that it indicated that at least the analysis in PANGEA confirmed sufficient power.

Thank you for your advice on the lack of information on Bayesian statistics. In our study, we used the Cauchy distribution with a default scale 0.707 in JASP that is analysis software. In accordance with your comments, we have added an explanation about the role of Bayesian analysis in these Results.

Although the effect shown in our overall results may not seem large, we consider that our findings fill the certain blanks that previous studies have not considered. So, we believe that it is meaningful to report this.

Comment 3: 

The expression “marginally significant” is inappropriate (page 11, line 176) as it does not reflect the arbitrary nature of the p-value. Furthermore, it appears that Authors use this label arbitrarily, as for example similar results (p=.08, eta=.15) are described as “marginally significant” (page 11, line 176) for the two-way interaction of interest and considered non-significant a few lines below (page 12, line 189). Thus, the section exploring the two-way interaction should be eliminated.

Related to the first point, the inclusion of correlations does not seem appropriate (at least with the separation by target type) because they follow results that appear unreliable (i.e., the interaction). Additionally, the correlations are not significant; thus, it is unclear what the value of Figure 4 is.

Response 3: We thank the reviewer for this comment. 

We also think that these points should be corrected to accommodate the concerns of this reviewer. Accordingly, we have rewritten the interpretation for the analysis in the Results. 

Moreover, we agree with your idea for Figure 4. Therefore, Figure 4 was removed from our paper.

Comment 4: 

The description of the methods of Experiment 1 is unclear, particularly on whether face validity and target valence target were manipulated independently. On first reading, I assumed that they were manipulated independently, which would be recommended. However, I reconsidered this assumption later on, when Experiment 2 was introduced. The Authors write “Second, because the valid-cue faces were predictive of both the location and valence of the targets, we could not determine which type of predictability was critical for the perceived trustworthiness to be modulated” (page 16, lines 253-255). This sentence suggests that valid-cue face always cued positive images, and this is a major issue casting doubts on the overall procedure of Experiment 1.

Response 4: 

We thank your insightful comments. Our writing was unclear. 

In Experiment 1, each face was associated with the same gaze pattern and the same valence of the targets during the gaze-cueing tasks. Previous studies have suggested that persons with consistent attitudes and behaviors tend to be evaluated as superior in the personality and intelligence. Therefore, it remains unclear whether the consistency of the gaze and the target types was critical for the perceived trustworthiness to be modulated. So, we examined whether such consistency influenced the facial trustworthiness by manipulating the predictability of the target valence. 

To help the interpretation for readers about the purpose in Experiment 2, we have added an explanation of this in the introduction in Experiment 2.

Comment 5: 

Experiment 2 substantially replicates the main effect of Experiment 1 and does not add much in its present form. Furthermore, the additional analyses (“Does history of target valence effect facial evaluation?” page 23, line 382) are poorly introduced/explained, and it is unclear what their contribution is, given the lack of clear and reliable results.

Response 5: We thank the reviewer for this comment. 

After reading comments 4 and 5, we felt that we had conveyed an unclear explanation about the purpose of Experiment 2. So, we rewrote relevant parts of the introduction of Experiment 2. 

In addition, following your comments, we also thought that we should explain the additional analysis more clearly. Accordingly, we have added the explanation to the Additional analysis.

---

## [Decision Letter · Decision Letter 1]

23 Oct 2019

PONE-D-19-14867R1

Affective evaluation of images influences personality judgments through gaze perception

PLOS ONE

Dear Dr. Shirai,

Thank you for submitting your manuscript to PLOS ONE. After careful consideration, we feel that it has merit but does not fully meet PLOS ONE’s publication criteria as it currently stands. Therefore, we invite you to submit a revised version of the manuscript that addresses the points raised during the review process.

We would appreciate receiving your revised manuscript by December 21, 2019. To enhance the reproducibility of your results, we recommend that if applicable you deposit your laboratory protocols in protocols.io, where a protocol can be assigned its own identifier (DOI) such that it can be cited independently in the future. For instructions see: http://journals.plos.org/plosone/s/submission-guidelines#loc-laboratory-protocols

We look forward to receiving your revised manuscript.

Kind regards,

Veronica Whitford, Ph.D.

Academic Editor

PLOS ONE

Reviewers' comments:

Reviewer's Responses to Questions

**Comments to the Author**

1. If the authors have adequately addressed your comments raised in a previous round of review and you feel that this manuscript is now acceptable for publication, you may indicate that here to bypass the “Comments to the Author” section, enter your conflict of interest statement in the “Confidential to Editor” section, and submit your "Accept" recommendation.

Reviewer #1: (No Response)

Reviewer #3: (No Response)

2. Is the manuscript technically sound, and do the data support the conclusions?

Reviewer #1: Partly

Reviewer #3: Partly

3. Has the statistical analysis been performed appropriately and rigorously? 

Reviewer #1: Yes

Reviewer #3: Yes

4. Have the authors made all data underlying the findings in their manuscript fully available?

Reviewer #1: Yes

Reviewer #3: No

5. Is the manuscript presented in an intelligible fashion and written in standard English?

Reviewer #1: Yes

Reviewer #3: Yes

6. Review Comments to the Author

Reviewer #1: 1) sample size: While the authors are correct in stating they cannot know the effect sizes for their study beforehand, they do have access to previous literature, and could thus calculate an effect size that would be more representative of standard cueing effects when manipulating trustworthiness, predictiveness and affective faces/targets. Opting for a “standard setting” is problematic when the authors don’t know why a particular program has a default listed

2) image numbers: thank you for the addition information

3) gaze photoediting: thank you for the addition information

4) methods details: thank you for the addition information. The manuscript continues to lack a Design section, however, which is a point raised in the first round of reviews. The Design section should list each of the factors (and their levels). It remains unclear whether the valid gaze cues were equally likely to look at positive and negative images, or whether these two points were confounded. Including details about the design within the methods section rather than as a small discussion point between the two experiments would clear up a lot of confusion. Further, it is unclear to me how the design of E1, where gaze-validity and target valence are confounded, will answer the authors main question of whether the emotional valence of a gazed object affects facial impression of the gazer, thus the rationale for including experiment 1 as it stands is unclear.

5) response-key mapping: thank you for the addition information

6) median RTs: the authors should include information about the distribution of the data and rationale for selecting medians in the manuscript proper.

7) figure axes: the choice of y-axis values is not that which I suggested, however it is preferable as compared to the original figure axes.

8) use of the term “marginal”: the authors have failed to remove the word “marginal” from both lines 181 and 193.

9) Thus twice. Fine now.

10) Design details: I am still unclear about the design of the study. Initially I believed that predictiveness and target valence were fully crossed, with 4 groups of faces. One group of faces always looked toward positive targets, the second group always looked at negative targets, the third group never looked at positive targets and the last group of faces never looked at negative targets. Then, when reading the other reviewer’s comments, it became clear that there is a confound between gaze validity and target valence, which means the data are confounded, and the design is not able to address the research question. After reading the revised section between E1 and E2, it is still unclear what the authors believe the first study adds to the literature.

11) trustworthiness ratings: how did the authors assess “whether there were the differences between two scores per face in Experiment 2.”, as they state in the letter to the reviewers? I suggest reporting the corresponding stats in the letter, or in the manuscript proper.

12) Figure 3 and [new] Figure 5: thank you for this addition

13) Raw data: thank you for including the raw data. The response times appear to be coded in seconds; the authors may want to either make a note of this somewhere, or convert to milliseconds to match their graphs

Taken together, while the authors have taken steps to improve the manuscript, there are still sections that are unclear, which makes it difficult to judge the technical details. It is also still not clear how E1 addresses the question the authors seek to answer, nor is it clear whether E1 is confounded - if it is, then the authors should consider removing that experiment.

Reviewer #3: This study presents a gaze-cueing paradigm extending and replicating an effect where faces that consistently look towards a target are trusted more than faces that consistently look away from the targets. In two experiments, this paper extends this established effect to judge how the valence of the target affects face evaluations, and reports that valid-cueing faces are only trusted more when they look towards affectively positive targets.

I like this study, and I think the question it addresses is well motivated and worthy of exploration and publication. However, while I think the paper is well written and well argued, I have some concerns regarding the manuscript as it currently stands. I have listed these below in order of their appearance through the manuscript.

Essentially, while I think there is reason to trust the primary finding – that trust learning is compromised for valid faces when they look at negative targets – my main concern is that I think there are many other points where the authors over-interpret weak or absent results. In particular, the additional analysis of Experiment 2 provides no conclusive evidence but is treated as support for various hypotheses. I think the paper could be greatly improved by drawing clearer parallels between the analysis stages and explicitly defined hypotheses and focussing the discussion to the key, substantiated findings.

• P.4 “For example, when we observe someone looking at a particular location, our attention automatically shifts to the same location (‘joint attention’)” – this reads like a definition of gaze-following, rather than joint attention.

• P.4 “Also, the viewer’s likability of the face was rated after the response to the target” – Regarding Bayliss & Tipper (2006), participants rated the trustworthiness of faces, not likability. This is an issue throughout the results section as well, where the authors describe their participants’ ratings as likability ratings when they were trustworthiness ratings. Care should be taken to be precise, particularly because this learning effect is not present when participants are asked how much they like the faces, only how much they trust them (cf. Strachan et al., 2016).

• P.6 “We investigated relatively more participants compared to the calculated sample size after considering the possibility of missing data” – the authors use the same rationale in Experiment 2 to justify an even larger sample size (and a much bigger increase relative to the power calculation). I see that the authors have responded to similar comments from the other reviewers but I am still confused. If missing data are such a concern then why is no missing or excluded data reported in Experiment 1? If there were no missing data in Experiment 1, why is it such a pressing concern in Experiment 2? Any missing data must be reported and justified.

• P.6. I am not convinced by the rationale for the correlation analyses. As I understand it, the prediction is that the participants who are more strongly affected by misleading gaze (and therefore experience greater RT costs) might be more likely to discriminate valid from invalid faces on the basis of trust. But I am not sure how this fits with the hypothesis that perception of a sender’s gaze alone would be sufficient to modulate trustworthiness without any attention reorienting. Evidence suggests that this trust learning is not driven by attention reorienting per se, but by a negative emotional reaction that is the result of misleading gaze (Manssuer et al., 2016). This emotional reaction (and the trust learning) do not appear to be related to the magnitude of the gaze-cueing cost, but by the interpretation of it.

• P.8-9. Details of the design are missing or incomplete. For example, “Since the pairs of a face and target were randomized for each trial” – this suggests that faces were paired equally often with any given target, and nowhere is it made explicit that faces were only paired with targets of a given valence. A dedicated paragraph on the experimental design could make this clearer. I note that another reviewer has already raised this point in a previous review (R1: Comment 4), but I still found the description unclear.

• I was not sure why no-cue faces that did not move their eyes were used – whether this was just as a control or an experimental condition with specific hypotheses. As I am not clear on why these faces were included in the design, this makes the findings much more complicated to interpret. Spell out clear justification, hypotheses and predictions for all conditions. This will make it much easier to see what contrasts are relevant to what.

• P.9 “a number from 1 to 9 appeared” should read “a scale from 1 to 9 appeared”

• P.9. A section in the Methods where the authors detail their data analysis plan and rationale (details of the frequentist ANOVA with post-hoc comparisons, Bayesian contrasts with clear explanation of what is tested and why, and the correlation analysis) would be helpful.

• P.11. “Furthermore, the two-way interaction was marginally significant”. Please drop the phrase ‘marginally significant’ throughout the manuscript – I believe both reviewers of the previous version also make this point.

• P.11-12. It is quite difficult to follow so many different analyses and understand which are justified follow-ups and which are not. Breaking up this whole results section into separate paragraphs and relating each analysis back to the specific hypothesis it is testing would make it easier to see which analyses are follow-up and which are conceptually separate.

• P.12 & P.22: “Mean of trustworthiness scores per conditions are illustrated in upper each bar.” Change to “Means of trustworthiness scores for each condition are shown above each bar”

• P.13. Line 214: Please always refer to a BF as either BF10 or BF01. Given that the authors talk about both in the text, it is important to be clear.

• P.13 & P.22. It is not clear if these Bayesian contrasts are directional or not. I think it is valid to form directional hypotheses on the basis of previous studies (e.g. that ratings to valid faces will be more positive than for invalid faces), but it is not clear at the moment what alternative hypothesis the authors are actually testing.

• P.14-15. Reiterate the rationale for the correlation analysis and the hypotheses it is testing. It would also be useful to explain how this finding ought to be interpreted, as it does not appear in the discussion section.

• P.18. See above comment re:p.6 regarding power calculations, sample size, and missing data. It is not clear what data might be missing when no filtering or exclusion criteria are reported. Please also report the reason for the increased sample size in the manuscript text.

• P.19. The use of “non-predictive” to refer to faces that consistently provide valid or invalid cues equally often to positive and negative images is confusing considering that this is a gaze cueing paradigm. All faces in Experiment 2 are predictive if one learns the contingencies, and they are predictive of the task-relevant feature (the target location). Non-predictive faces, such as were used in the original Bayliss & Tipper design, would be faces that provide 50/50 valid and invalid cues. As the authors are referring to a different kind of contingency, I would advise using a different term throughout to avoid confusion (maybe ‘inconsistent’ in their preference for positive or negative targets?).

• P.19. Using 4 types of faces in a within-subjects design is quite complicated for this kind of learning task, and I suspect that not only is the complicated design working against the authors in terms of interpreting their results, but also making it much more difficult for participants to keep track of face-gaze-valence associations. This limitation should be acknowledged, and care taken to avoid over-interpreting spurious results.

• P.20. There were two ratings for each face that were then averaged. Please provide a measure of consistency or reliability across ratings. The authors replied to a comment from the reviewers in the previous round that said that these ratings were consistent, but I would like to see this in the manuscript (a footnote would suffice).

• P.23. “Moreover, BF10 in the condition comparing positive and neutral valence-cue was 2.62, indicating that positive emotional valence of gaze at targets increased facial trustworthiness.” This Bayes factor indicates anecdotal evidence, and so the support for this interpretation is weak at best.

• P.25. “Because the non-predictive faces looked at the positive or the negative target with equal probability, we can assume that the perceived trustworthiness of the non-predictive faces were not systematically modulated by repetitive presentations in the gaze-cue phase.” Not necessarily. Learning about faces may be biased towards cheater detection and learning about invalid faces may be prioritised as a result (Bell et al., 2012; Buchner et al., 2009; Strachan & Tipper, 2017).

• P.25. The data used for this analysis do not seem to be in the Supplementary Material.

• P.25. All of the Bayes factors here indicate very weak evidence either way, and this analysis provides no conclusive answer to any of the contrasts described.

• P.27. “However, we cannot disregard the possibility that the emotional information in the images boosted trustworthiness or enhanced another positive affect such as attractiveness. In future research, we should examine the types of impressions that are influenced by the emotional valence of gazed images or events.” With regards to directionality, the continuous pre- and post-gaze-cueing continuous trust ratings introduced in Manssuer et al. (2015) allow for easier tracking of whether this learning effect is driven by an increase in trust for valid faces or a decrease in trust for invalid faces relative to baseline. This could be worth considering when thinking about future research.

• P.27. Line 441-446. This interpretation is based on very weak evidence and inconclusive Bayesian analysis. I do not feel that these conclusions are justified.

• A general point regarding the effect that the authors are investigating: since the original Bayliss & Tipper (2006) paper, there have been several studies from Steven Tipper’s lab that have replicated and extended the trust learning effect, and these form the bulk of the literature on this effect. I have mentioned a few in this review (including a couple of my own). However, one that I think is of particular interest and should be discussed is Kirkham et al. (2015) PLOS One. “Facial Mimicry and Emotion Consistency: Influences of Memory and Context.” This addresses a very similar question to the current study but uses overt expressions of emotion to manipulate trust and find that faces that show inconsistent expressions (i.e. smiling at negative targets or frowning at positive targets) are distrusted. Embedding the current study in the context of existing literature and interpreting the findings in light of previous work will greatly strengthen the theoretical contribution of the paper.

7. PLOS authors have the option to publish the peer review history of their article (what does this mean?). If published, this will include your full peer review and any attached files.

Reviewer #1: No

Reviewer #3: Yes: James Strachan

---

## [Author Response · Author response to Decision Letter 1]

18 Dec 2019

December 19th, 2019

Reference: PONE-D-19-14867

Resubmission

“Affective evaluation of images influences personality judgments through gaze perception”

We appreciate for your valuable comments on our paper. We have revised the manuscript by following your comments. Our responses to the reviewers’ comments are below.

RESPONSE TO REVIEWER 1:

Comment 1: 

sample size: While the authors are correct in stating they cannot know the effect sizes for their study beforehand, they do have access to previous literature, and could thus calculate an effect size that would be more representative of standard cueing effects when manipulating trustworthiness, predictiveness and affective faces/targets. Opting for a “standard setting” is problematic when the authors don’t know why a particular program has a default listed

Response 1: Thank you for the comment. 

As you stated, there have been similar studies, however, since our study and the previous study did not have the same experimental design, we did not refer to the effect sizes of the previous studies. 

Also, we feel that the expression "standard settings" may have caused a misunderstanding. In order to avoid large power overflows and too little power, we calculated the sample size by using the medium effect size (f).

Comment 2: 

methods details: thank you for the addition information. The manuscript continues to lack a Design section, however, which is a point raised in the first round of reviews. The Design section should list each of the factors (and their levels). It remains unclear whether the valid gaze cues were equally likely to look at positive and negative images, or whether these two points were confounded. Including details about the design within the methods section rather than as a small discussion point between the two experiments would clear up a lot of confusion. Further, it is unclear to me how the design of E1, where gaze-validity and target valence are confounded, will answer the authors main question of whether the emotional valence of a gazed object affects facial impression of the gazer, thus the rationale for including experiment 1 as it stands is unclear.

Response 2: 

Thank you for the comment. 

We agree with the reviewer’s comment. According to the comment, we have included the details of the experimental designs in the Design section.

Comment 3: 

median RTs: the authors should include information about the distribution of the data and rationale for selecting medians in the manuscript proper.

Response 3: We thank the reviewer for this comment. 

We have included information about the distribution. Since it was not a normal distribution, we used the median.

Comment 4: 

use of the term “marginal”: the authors have failed to remove the word “marginal” from both lines 181 and 193.

Response 4: Thank you for this comment. We have removed the word “marginal”.

Comment 5:

Design details: I am still unclear about the design of the study. Initially I believed that predictiveness and target valence were fully crossed, with 4 groups of faces. One group of faces always looked toward positive targets, the second group always looked at negative targets, the third group never looked at positive targets and the last group of faces never looked at negative targets. Then, when reading the other reviewer’s comments, it became clear that there is a confound between gaze validity and target valence, which means the data are confounded, and the design is not able to address the research question. After reading the revised section between E1 and E2, it is still unclear what the authors believe the first study adds to the literature.

Response 5: We welcome this insightful comment. We should have explained the design of our study more clearly. Therefore, we have included an explanation about the experimental settings in the Design section. Moreover, to assist the interpretations by the readers, we also have included a Figure in the Design section.

Comment 6:

trustworthiness ratings: how did the authors assess “whether there were the differences between two scores per face in Experiment 2.”, as they state in the letter to the reviewers? I suggest reporting the corresponding stats in the letter, or in the manuscript proper.

Response 6: We thank you for your comments. We have included these results in a footnote of the Results section (p.24).

Comment 7: 

Raw data: thank you for including the raw data. The response times appear to be coded in seconds; the authors may want to either make a note of this somewhere, or convert to milliseconds to match their graphs

Response 7: We appreciate your comment on this point. We have included an explanation about the response times in S2_Appendix.

RESPONSE TO REVIEWER 2:

Comment 1: 

P.4 “For example, when we observe someone looking at a particular location, our attention automatically shifts to the same location (‘joint attention’)” – this reads like a definition of gaze-following, rather than joint attention.

Response 1: Thank you for your insightful comments. We have revised this point accordingly (p.4-5).

Comment 2: 

 P.4 “Also, the viewer’s likability of the face was rated after the response to the target” – Regarding Bayliss & Tipper (2006), participants rated the trustworthiness of faces, not likability. This is an issue throughout the results section as well, where the authors describe their participants’ ratings as likability ratings when they were trustworthiness ratings. Care should be taken to be precise, particularly because this learning effect is not present when participants are asked how much they like the faces, only how much they trust them (cf. Strachan et al., 2016).

Response 2: We express our deep appreciation to this reviewer for this insightful comment. In accordance with your comments, we have revised the explanation about previous studies (p.4, line 56-59). 

Comment 3: 

 P.6 “We investigated relatively more participants compared to the calculated sample size after considering the possibility of missing data” – the authors use the same rationale in Experiment 2 to justify an even larger sample size (and a much bigger increase relative to the power calculation). I see that the authors have responded to similar comments from the other reviewers but I am still confused. If missing data are such a concern then why is no missing or excluded data reported in Experiment 1? If there were no missing data in Experiment 1, why is it such a pressing concern in Experiment 2? Any missing data must be reported and justified.

Comment 15: 

P.18. See above comment re:p.6 regarding power calculations, sample size, and missing data. It is not clear what data might be missing when no filtering or exclusion criteria are reported. Please also report the reason for the increased sample size in the manuscript text.

Response 3 and 15: We thank the reviewer for this comment. There were no missing data in our experiments. At times, certain participants interrupted the experiment of another study, and therefore, we attempted to recruit a slightly higher number of participants for all experiments.

Also, there was a delay in stopping the recruitment on the Internet when the required number of people were collected, which resulted in a slightly large number of participants in Experiment 2. 

Our discussion is not only based on the results of ANOVAs, but also the Bayesian analysis. Moreover, the effect shown in our overall results is not large. Therefore, we believe that the number of participants in our study did not strongly influence the results of our study.

Comment 4: 

 P.6. I am not convinced by the rationale for the correlation analyses. As I understand it, the prediction is that the participants who are more strongly affected by misleading gaze (and therefore experience greater RT costs) might be more likely to discriminate valid from invalid faces on the basis of trust. But I am not sure how this fits with the hypothesis that perception of a sender’s gaze alone would be sufficient to modulate trustworthiness without any attention reorienting. Evidence suggests that this trust learning is not driven by attention reorienting per se, but by a negative emotional reaction that is the result of misleading gaze (Manssuer et al., 2016). This emotional reaction (and the trust learning) do not appear to be related to the magnitude of the gaze-cueing cost, but by the interpretation of it.

Response 4: We thank for your insightful comments. We are sorry that our writing caused a misreading. As you have mentioned, previous studies have shown that attentional orientation does not appear to be related to the results of the facial evaluation. We conducted a correlation analysis because we wanted to confirm whether our results were consistent with the results of previous studies (i.e., whether the boosted facial evaluation was explained by the observer's attentional orienting).

We have revised the explanation of the hypothesis in the introduction (p.6, line 87-92).

Comment 5: 

 P.8-9. Details of the design are missing or incomplete. For example, “Since the pairs of a face and target were randomized for each trial” – this suggests that faces were paired equally often with any given target, and nowhere is it made explicit that faces were only paired with targets of a given valence. A dedicated paragraph on the experimental design could make this clearer. I note that another reviewer has already raised this point in a previous review (R1: Comment 4), but I still found the description unclear.

Comment 6: 

I was not sure why no-cue faces that did not move their eyes were used – whether this was just as a control or an experimental condition with specific hypotheses. As I am not clear on why these faces were included in the design, this makes the findings much more complicated to interpret. Spell out clear justification, hypotheses and predictions for all conditions. This will make it much easier to see what contrasts are relevant to what.

Response 5 and 6: We thank the reviewer for this comment. 

According to your comments, we have more clearly explained the experimental design (including the explanation about no-cue faces in the Design sections of Experiments 1 and 2. 

Comment 7: 

P.9 “a number from 1 to 9 appeared” should read “a scale from 1 to 9 appeared”

Response 7: Thank you for your comments. We have revised this point (p.9, line 144).

Comment 8: 

P.9. A section in the Methods where the authors detail their data analysis plan and rationale (details of the frequentist ANOVA with post-hoc comparisons, Bayesian contrasts with clear explanation of what is tested and why, and the correlation analysis) would be helpful.

Comment 10:

P.11-12. It is quite difficult to follow so many different analyses and understand which are justified follow-ups and which are not. Breaking up this whole results section into separate paragraphs and relating each analysis back to the specific hypothesis it is testing would make it easier to see which analyses are follow-up and which are conceptually separate.

Response 8 and 10: We welcome this insightful comment. As you have stated, we also felt that separating the results by type of analysis obscures the understanding of the purpose and hypothesis of each analysis. Therefore, we changed the layout of the results. 

Comment 9:

P.11. “Furthermore, the two-way interaction was marginally significant”. Please drop the phrase ‘marginally significant’ throughout the manuscript – I believe both reviewers of the previous version also make this point.

Response 9: We thank the reviewer for this comment. We have removed this word (p. 13).

Comment 11: 

P.12 & P.22: “Mean of trustworthiness scores per conditions are illustrated in upper each bar.” Change to “Means of trustworthiness scores for each condition are shown above each bar”

Response 11: We thank the reviewer for this comment. We have revised this point.

Comment 12: 

P.13. Line 214: Please always refer to a BF as either BF10 or BF01. Given that the authors talk about both in the text, it is important to be clear.

Response 12: We thank the reviewer for this comment. We have added information about the Bayes factors. 

Comment 13:

P.13 & P.22. It is not clear if these Bayesian contrasts are directional or not. I think it is valid to form directional hypotheses on the basis of previous studies (e.g. that ratings to valid faces will be more positive than for invalid faces), but it is not clear at the moment what alternative hypothesis the authors are actually testing.

Response 13: We thank the reviewer for this comment. We agree with your comments. To clarify the hypothesis, we have added information about the hypothesis of Bayesian analysis. 

Comment 14: 

 P.14-15. Reiterate the rationale for the correlation analysis and the hypotheses it is testing. It would also be useful to explain how this finding ought to be interpreted, as it does not appear in the discussion section.

Response 14: We thank the reviewer for this comment. According to your comment, we have added the rationale and hypothesis for the correlation analysis.

Comment 16: 

P.19. The use of “non-predictive” to refer to faces that consistently provide valid or invalid cues equally often to positive and negative images is confusing considering that this is a gaze cueing paradigm. All faces in Experiment 2 are predictive if one learns the contingencies, and they are predictive of the task-relevant feature (the target location). Non-predictive faces, such as were used in the original Bayliss & Tipper design, would be faces that provide 50/50 valid and invalid cues. As the authors are referring to a different kind of contingency, I would advise using a different term throughout to avoid confusion (maybe ‘inconsistent’ in their preference for positive or negative targets?).

Response 16: We appreciate the reviewer for this comment. As you have stated, we also thought that the expression “non-predictive” confounded the readers’ understanding. Therefore, we changed the name of the condition from “non-predictive” to “random”.

Comment 17: 

P.19. Using 4 types of faces in a within-subjects design is quite complicated for this kind of learning task, and I suspect that not only is the complicated design working against the authors in terms of interpreting their results, but also making it much more difficult for participants to keep track of face-gaze-valence associations. This limitation should be acknowledged, and care taken to avoid over-interpreting spurious results.

Response 17: We thank the reviewer for this comment. 

We agree with your idea and also consider that this possibility should be carefully discussed (p. 33, lines 528-529). 

Comment 18: 

P.20. There were two ratings for each face that were then averaged. Please provide a measure of consistency or reliability across ratings. The authors replied to a comment from the reviewers in the previous round that said that these ratings were consistent, but I would like to see this in the manuscript (a footnote would suffice).

Response 18: We thank the reviewer for this insightful comment. We have added these results to a footnote in the Results section (p. 24).

Comment 19: 

P.23. “Moreover, BF10 in the condition comparing positive and neutral valence-cue was 2.62, indicating that positive emotional valence of gaze at targets increased facial trustworthiness.” This Bayes factor indicates anecdotal evidence, and so the support for this interpretation is weak at best.

Comment 22: 

P.25. All of the Bayes factors here indicate very weak evidence either way, and this analysis provides no conclusive answer to any of the contrasts described.

Response 19 and 22: We thank the reviewer for this comment. To discuss this more carefully, we have changed the expression of the interpretation of these results (p. 26, 28-29).

Comment 20: 

P.25. “Because the non-predictive faces looked at the positive or the negative target with equal probability, we can assume that the perceived trustworthiness of the non-predictive faces were not systematically modulated by repetitive presentations in the gaze-cue phase.” Not necessarily. Learning about faces may be biased towards cheater detection and learning about invalid faces may be prioritised as a result (Bell et al., 2012; Buchner et al., 2009; Strachan & Tipper, 2017).

Response 20: We thank the reviewer for this comment. We are sorry that our explanation was insufficient and has caused a misunderstanding. To clarify this, we have revised the prediction of the results (p. 28). 

Comment 21: 

P.25. The data used for this analysis do not seem to be in the Supplementary Material.

Response 21: We thank the reviewer for this comment. We have included all the data to Supplementary Materials.

Comment 23: 

P.27. “However, we cannot disregard the possibility that the emotional information in the images boosted trustworthiness or enhanced another positive affect such as attractiveness. In future research, we should examine the types of impressions that are influenced by the emotional valence of gazed images or events.” With regards to directionality, the continuous pre- and post-gaze-cueing continuous trust ratings introduced in Manssuer et al. (2015) allow for easier tracking of whether this learning effect is driven by an increase in trust for valid faces or a decrease in trust for invalid faces relative to baseline. This could be worth considering when thinking about future research.

Response 23: We deeply appreciate the reviewer for this comment. We want to consider this point in future studies. 

Comment 24: 

P.27. Line 441-446. This interpretation is based on very weak evidence and inconclusive Bayesian analysis. I do not feel that these conclusions are justified.

Response 24: We thank the reviewer for this comment. 

According to your comment, we have included a discussion about this point (p. 29 line 460-464).

Comment 25: 

A general point regarding the effect that the authors are investigating: since the original Bayliss & Tipper (2006) paper, there have been several studies from Steven Tipper’s lab that have replicated and extended the trust learning effect, and these form the bulk of the literature on this effect. I have mentioned a few in this review (including a couple of my own). However, one that I think is of particular interest and should be discussed is Kirkham et al. (2015) PLOS One. “Facial Mimicry and Emotion Consistency: Influences of Memory and Context.” This addresses a very similar question to the current study but uses overt expressions of emotion to manipulate trust and find that faces that show inconsistent expressions (i.e. smiling at negative targets or frowning at positive targets) are distrusted. Embedding the current study in the context of existing literature and interpreting the findings in light of previous work will greatly strengthen the theoretical contribution of the paper.

Response 25: We highly appreciate the reviewer for this insightful comment. As you have stated, we have attempted to embed our findings in the literature of previous studies (such as Kirkham et al., 2015), and then explain the value of the present study. Thus, we have revised the General discussion.

---

## [Decision Letter · Decision Letter 2]

12 Feb 2020

PONE-D-19-14867R2

Affective evaluation of images influences personality judgments through gaze perception

PLOS ONE

Dear Dr. Shirai,

Thank you for submitting your manuscript to PLOS ONE. As you will see below, the reviewers provided insightful and detailed comments on your revised manuscript. They were generally satisfied with your revisions; however, they have some important lingering concerns. These include the clarity of some of the descriptions in your Introduction and Methods sections, the lack of sample size calculations, and insufficient information on the analyses you performed (and interpretation thereof). These concerns preclude the publication of your paper at this point; however, I invite you to submit a revised version that addresses each of the reviewers' points.

We would appreciate receiving your revised manuscript by April 13, 2020. To enhance the reproducibility of your results, we recommend that if applicable you deposit your laboratory protocols in protocols.io, where a protocol can be assigned its own identifier (DOI) such that it can be cited independently in the future. For instructions see: http://journals.plos.org/plosone/s/submission-guidelines#loc-laboratory-protocols

We look forward to receiving your revised manuscript.

Kind regards,

Veronica Whitford, Ph.D.

Academic Editor

PLOS ONE

Reviewers' comments:

Reviewer's Responses to Questions

**Comments to the Author**

1. If the authors have adequately addressed your comments raised in a previous round of review and you feel that this manuscript is now acceptable for publication, you may indicate that here to bypass the “Comments to the Author” section, enter your conflict of interest statement in the “Confidential to Editor” section, and submit your "Accept" recommendation.

Reviewer #1: (No Response)

Reviewer #3: (No Response)

2. Is the manuscript technically sound, and do the data support the conclusions?

Reviewer #1: Partly

Reviewer #3: Yes

3. Has the statistical analysis been performed appropriately and rigorously? 

Reviewer #1: Yes

Reviewer #3: Yes

4. Have the authors made all data underlying the findings in their manuscript fully available?

Reviewer #1: Yes

Reviewer #3: Yes

5. Is the manuscript presented in an intelligible fashion and written in standard English?

Reviewer #1: Yes

Reviewer #3: Yes

6. Review Comments to the Author

Reviewer #1: 1) Beginning line 87: “If facial trustworthiness requires attentional orienting…” This is very unclear to me. What are you trying to say? Based on Bayliss & Tipper (2006), one would hypothesize that the magnitude of the cueing effect would not be modulated by trustworthiness, but rather just the ratings. I also don’t understand the predicted directionality here. Why would changes in facial trustworthiness depend on the strength of the gaze cueing effect? As mentioned above, Bayliss and Tipper found a dissociation between attention and trust judgments. This is repeated again beginning line 262, and is still confusing.

2) What would you predict for the invalid-cue faces?

3) sample size: At this point it is clear that the authors are not willing to conduct sample size calculations based on REAL previous data, as I have requested twice now. While I am clearly not satisfied, I will leave it to the AE to make the final decision here.

3) Design should come before Procedure

4) lines 134-136, beginning: “Since the pairs of a face and …”. This feels misleading – if I understand the design correctly, each face identity is paired with a certain valence of target image. While it is strictly true that face identities did not predict a particular image, they did predict a subset of those images, thus I would take care to be very clear here.

5) Methods: “Therefore, the faces were randomly assigned to gender-matched six groups of six faces. Each face was always paired with a target having the same valence.” (lines 155-157). The wording is confusing. Why not clearly write the design here? For example, “For each participant, one face was assigned to each experimental cell, such that one identity always looked at the positive targets (e.g., valid-cue, positive target), another identity never looked at the positive targets (invalid-cue, positive target), and so on. Across participants, the assignment of face identity and condition was randomized.”

6) Figure captions: the figure caption for the new Figure 2 does not provide any helpful information. I suggest explaining what is going on in the figure.

7) In the positive target condition, are participants actually faster to response for valid cues as compared to invalid cues? The graph suggests there is no difference here, and in fact it is the difference in the negative target condition that may be driving the significant follow-up t-test run on lines 185-186. If the gaze cueing effect is not significantly different in the positive target condition, then this calls into question some of the conclusions.

8) While the authors removed the term “marginal” from their analyses, they merely replaced it with a similar sentiment (e.g., “reasonably close to significant”). This should be removed. Along similar lines, once the authors state at lines 206-207 that the two-way interaction was not significant, they should not continue to conduct further analyses. The authors should remove lines 207-225, unless they can show a compelling reason for conduct specific targeted analyses due to a priori predictions.

9) Experiment 2: shouldn’t the factor be “valence-target” rather than “valence-cue”?

10) E2: what would you predict for valid positive versus valid neutral?

11) The authors state that BF10 >3 provides moderate support for the null, and a BF10>10 provides strong support for the alternative hypothesis. However, when interpreting the Bayesian analyses in Experiment 2, the authors do not adhere to these values. There is some support for the null in 4 comparisons (Valid: Neutral vs Random; Invalid: Positive vs Neutral, Positive vs Random, and Neutral vs Random), and no support for the alternative. The authors should not attempt to interpret the BF values that do not meet the threshold.

12) For the additional analysis beginning line 440, the BF do not weakly support the null, as the values do not reach 3.

13) The results from the correlation analyses converge with those of Bayliss and Tipper (2006), which should be reported.

14) E2 discussion: what is the evidence that “negative targets decreased the evaluation for valid position-cue faces” (lines 476-477). For the evaluation ANOVA, while there was a main effect of validity and main effect of target rating, there was no interaction. Please point the readers to the correct analyses to bring about this conclusion

Reviewer #3: The authors have addressed most of my comments on the previous version of the manuscript adequately. The manuscript is much improved and reads much more clearly. I have a few outstanding comments

• I think the change from ‘non-predictive’ to ‘random’ in Experiment 2 is a good one. The authors should also change the column headings in the supporting information data file for consistency.

• P.6, line 90. “if increased facial evaluation did not require attentional orienting by the viewer” – the authors could describe what would be a likely explanation other than attention reorienting (I would think the experience of joint attention on a positive or negative object

• P.7, line 114. It is still wholly not clear how many face identities were picked. Were the stimuli images of 36 different individuals, or 36 images of 12 individuals? From elsewhere I take it as the former, but this should be clear here.

• P.9, line 147. “The faces and the targets in Fig 1 are not the original images that were used in our study, but similar images used for illustrative purposes” – this implies that face images used were line drawings, as in Figure 1. I would remove the word “similar” here and in the legend of Figure 2 on p.11, line 170, or replace it with “schematic” or something similar.

• P.11, line 179. Report the results of the test of normality.

• P.13, line 206 & 218. “reasonably close to significant” is no better than “marginally significant”. It is fine to draw attention to differences or trends, but these should not be oversold. Describe these as non-significant.

• P.14, line 220. “we found the effect of face type, F(1, 19) = 3.48, p = .08, ηp2 = .15” – this is not significant, and so post-hoc tests are not justified.

• P.16, line 257. Change “likeability” to “trustworthiness”. See also lines 262, 264, 267, 279, 320, 467-8, & 535.

• P.16, line 260. This correlation analysis does not “assess whether attentional orienting was required for increased facial trustworthiness”, as most people would show attentional orienting in response to gaze cues. If trustworthiness is driven by this reorienting you could hypothesis that the magnitude of this attentional reorienting would show a relationship to the trust learning, but this is not the same as testing whether attentional reorienting is required. Gaze following could well be necessary for this effect without there necessarily being a linear relationship between gaze following and trust learning.

• P.27, line 440. “Additional analysis: did evaluation scores influence memory traces of the relationships between face and image?” This title does not match the logic of the analysis described. The described analysis does not test the effect of evaluation on memory, but instead explores whether the results could be explained as a response to the single preceding trial rather than the result of iterative learning. Reword this title to accurately reflect the analysis described.

• P.30, line 480. The consistency of the target valence may not affect face evaluation, but all faces were consistent in their behaviour (looking towards or away from the target) in Experiment 2. This consistency could win out over the fairly minimal target valence manipulation.

• P.31, lines 505-508. The lead into describing Kirkham et al is bit weak, and I am not convinced that the weak evidence for learned representations in Experiment 2 fits with the results of that study. Instead, I would rewrite this section, using Kirkham et al. as evidence that participants learn to trust and distrust people based on what they look at and how they respond. A valuable contribution of the current study is that extends this result and shows that, even without an emotional reaction, what people consistently attend can affect how trustworthy they are perceived to be.

• P.32, line 529. “paring” should be “pairing”

7. PLOS authors have the option to publish the peer review history of their article (what does this mean?). If published, this will include your full peer review and any attached files.

Reviewer #1: No

Reviewer #3: No

---

## [Author Response · Author response to Decision Letter 2]

10 Apr 2020

RESPONSE TO REVIEWER 1:

Comment 1: 

Beginning line 87: “If facial trustworthiness requires attentional orienting…” This is very unclear to me. What are you trying to say? Based on Bayliss & Tipper (2006), one would hypothesize that the magnitude of the cueing effect would not be modulated by trustworthiness, but rather just the ratings. I also don’t understand the predicted directionality here. Why would changes in facial trustworthiness depend on the strength of the gaze cueing effect? As mentioned above, Bayliss and Tipper found a dissociation between attention and trust judgments. This is repeated again beginning line 262, and is still confusing.

Response 1: Thank you for this comment. As you have stated, this sentence was not clear. Therefore, we have included the following revision in Line 89 to clarify our meaning, “Furthermore, there is a controversy regarding the source of modulation of perceived trustworthiness based on gaze cue: Specific studies have suggested that the increased likability of faces or objects might be due to enhanced perceptual fluency caused by attentional facilitation elicited by gaze cues ([8, 31]). However, others have reported no correlation between changes in likability and the degree of cueing effects ([10, 13, 27]). Therefore, we examined the relationship between changes in perceived facial trustworthiness and the degree of the gaze cueing effect.”.

Comment 2: 

What would you predict for the invalid-cue faces?

Response 2: 

Thank you for the comment. According to your comments, we added the following text to Line 87, “By contrast, such changes in facial evaluation would not be observed when faces consistently looked away from emotional targets.”.

Comment 3: 

sample size: At this point it is clear that the authors are not willing to conduct sample size calculations based on REAL previous data, as I have requested twice now. While I am clearly not satisfied, I will leave it to the AE to make the final decision here.

Response 3: We thank the reviewer for the conscientious comments. We also consider that the sample size calculation should be based on real data from similar studies. However, to our knowledge, no previous studies have been conducted on the evaluation of faces by gaze direction with the same design as this study. As a result, we could not refer to the effect sizes of previous studies. In such a case, we believe that calculating the sample size to ensure sufficient power using medium effect sizes for avoiding large power overflows or low power, was not the inappropriate procedure. Therefore, based on Cohen (1992), we decided to use the medium effect size for sample size calculation. We have explained this in Materials and methods. We hope that this explanation will clarify our previous ambiguous response.

Comment 4: 

Design should come before Procedure

Response 4: Thank you for this comment. We have moved the Design section above the Procedure section.

Comment 5: lines 134-136, beginning: “Since the pairs of a face and …”. This feels misleading – if I understand the design correctly, each face identity is paired with a certain valence of target image. While it is strictly true that face identities did not predict a particular image, they did predict a subset of those images, thus I would take care to be very clear here.

Response 5: We welcome this insightful comment. We have included the following sentences in the Procedure section to clarify this point (Lines 144-146), “Since the pairs of faces and targets were randomized for each trial, the face identities did not predict if a particular image would appear as the target, whereas they predicted the valence of those images.”

Comment 6: Methods: “Therefore, the faces were randomly assigned to gender-matched six groups of six faces. Each face was always paired with a target having the same valence.” (lines 155-157). The wording is confusing. Why not clearly write the design here? For example, “For each participant, one face was assigned to each experimental cell, such that one identity always looked at the positive targets (e.g., valid-cue, positive target), another identity never looked at the positive targets (invalid-cue, positive target), and so on. Across participants, the assignment of face identity and condition was randomized.”

Response 6: We thank you for your comments. Based on your comment, we have included the following sentences in the Design section to clarify the design of the current study (Lines 140-144). “The faces were assigned to each experimental cell for each participant, such that some faces consistently looked at positive targets (e.g., valid-cue, positive target), whereas other faces never looked at positive targets (invalid-cue, positive target), and so on. Each experimental cell contained six faces, and the gender ratios of these faces were consistent across each experimental cell.

Comment 7: Figure captions: the figure caption for the new Figure 2 does not provide any helpful information. I suggest explaining what is going on in the figure.

Response 7: We appreciate your comment on this point. We have added the following explanation to the caption of Figure 2, “The face-target pairs used in Experiment 1 are surrounded by the dotted line, and those in Experiment 2 are surrounded by the solid line. The vertical part indicates the type of face, and the horizontal part indicates the type of target. In Experiment 1, there were 6 pairs of faces (valid-cue, invalid-cue, no-cue) and targets (positive, negative). In Experiment 2, there were 8 pairs of faces (valid-cue, invalid-cue) and targets (positive, negative, neutral, random).

Comment 8: In the positive target condition, are participants actually faster to response for valid cues as compared to invalid cues? The graph suggests there is no difference here, and in fact it is the difference in the negative target condition that may be driving the significant follow-up t-test run on lines 185-186. If the gaze cueing effect is not significantly different in the positive target condition, then this calls into question some of the conclusions.

Response 8: We appreciate your comment on this point and we agree with your comment. As you have stated, we can observe from Figure 3 that the gaze cueing effect was greater for positive targets than for negative targets. However, the interaction between the face type and the target type was not statistically significant (Line 206). Even if the target type affected the gaze cueing effect, since there were no correlation between the gaze cueing effect and increased facial trustworthiness (line 259), we think that it is difficult to consider that the results of facial evaluations are strongly changed by differential magnitudes of gaze-cueing effect in target types.

Comment 9: While the authors removed the term “marginal” from their analyses, they merely replaced it with a similar sentiment (e.g., “reasonably close to significant”). This should be removed. Along similar lines, once the authors state at lines 206-207 that the two-way interaction was not significant, they should not continue to conduct further analyses. The authors should remove lines 207-225, unless they can show a compelling reason for conduct specific targeted analyses due to a priori predictions.

Response 9: We appreciate your comment on this point. We have removed the results of further analysis.

Comment 10: Experiment 2: shouldn’t the factor be “valence-target” rather than “valence-cue”?

Response 10: We appreciate your comment on this point. We have changed the phrase “valence-cue” to “valence-target”.

Comment 11: E2: what would you predict for valid positive versus valid neutral?

Response 11: We appreciate your comment on this point. We predicted that the faces that always looked at positive targets were evaluated as more trustworthy than the faces that always looked at the neutral targets. 

Moreover, both the Introduction and the Design in Experiment 2 described the prediction for results. Therefore, we integrated this information in the Introduction on Lines 292-295.

Comment 12: The authors state that BF10 >3 provides moderate support for the null, and a BF10>10 provides strong support for the alternative hypothesis. However, when interpreting the Bayesian analyses in Experiment 2, the authors do not adhere to these values. There is some support for the null in 4 comparisons (Valid: Neutral vs Random; Invalid: Positive vs Neutral, Positive vs Random, and Neutral vs Random), and no support for the alternative. The authors should not attempt to interpret the BF values that do not meet the threshold.

Comment 13: For the additional analysis beginning line 440, the BF do not weakly support the null, as the values do not reach 3.

Response 12 and 13: We appreciate your comment on this point. According to your comment, we have added the following sentences on Lines 406-409 about the need for caution when discussing BF values, “The BF10 in the comparison between positive and neutral valence-targets and between negative and neutral valence-targets was smaller than 3, which indicates weak evidence. Therefore, we could not firmly conclude that there was an effect of the emotional valence of each target on the facial evaluation”. We also revised the text on Lines 436-442, “The BF10 supporting the hypothesis that valid faces in the positive condition were rated as more trustworthy than valid faces in the random condition looking at positive targets was slightly higher than the BF01, which supported the null hypothesis (BF10 = 1.44, BF01 = 0.70). Furthermore, BF10 supporting the hypothesis that valid faces in the negative condition were evaluated as more untrustworthy than valid faces in the random condition looking at negative targets was also slightly higher than BF01, which supported the null hypothesis (BF10 = 1.77, BF01 = 0.56)”.

Comment 14: The results from the correlation analyses converge with those of Bayliss and Tipper (2006), which should be reported.

Response 14: We appreciate your comment on this point, and we agree with you. We added the information about the previous evidence on Lines 90-92.

Comment 15: E2 discussion: what is the evidence that “negative targets decreased the evaluation for valid position-cue faces” (lines 476-477). For the evaluation ANOVA, while there was a main effect of validity and main effect of target rating, there was no interaction. Please point the readers to the correct analyses to bring about this conclusion

Response 15: We appreciate your comment on this point. To clarify this point, we have included the following texts on Lines 459-462 of Discussion, “Moreover, valid-cue faces with neutral targets were regarded as more trustworthy than valid-cue faces with negative targets, which were less trustworthy than valid-cue faces with positive targets. The Bayesian analyses supported these tendencies, suggesting that both emotional valences impacted facial trustworthiness.”

RESPONSE TO REVIEWER 2:

Comment 1: I think the change from ‘non-predictive’ to ‘random’ in Experiment 2 is a good one. The authors should also change the column headings in the supporting information data file for consistency.

Response 1: Thank you for your insightful comments. We have changed from “non-predictive” to “random”.

Comment 2: 

 P.6, line 90. “if increased facial evaluation did not require attentional orienting by the viewer” – the authors could describe what would be a likely explanation other than attention reorienting (I would think the experience of joint attention on a positive or negative object

Response 2: We express our deep appreciation for this insightful comment. To clarify the prediction of the results, we have rewritten the sentences on Line 89, “Furthermore, there is a controversy regarding the source of modulation of perceived trustworthiness based on gaze cue: Specific studies have suggested that the increased likability of faces or objects might be due to enhanced perceptual fluency caused by attentional facilitation elicited by gaze cues ([8, 31]). However, others have reported no correlation between changes in likability and the degree of cueing effects ([10, 13, 27]). Therefore, we examined the relationship between changes in perceived facial trustworthiness and the degree of the gaze cueing effect.” Moreover, we added the discussion of the explanation other than the attentional orienting on Lines 523-525 in the General discussion.

Comment 3: P.7, line 114. It is still wholly not clear how many face identities were picked. Were the stimuli images of 36 different individuals, or 36 images of 12 individuals? From elsewhere I take it as the former, but this should be clear here.

Response 3: We thank the reviewer for this comment. The former statement (i.e., the stimuli images of 36 different individuals) is correct. To clarify this point, we have rewritten the text on Line 142, “The face images of 36 different individuals were randomly assigned to each experimental cell for each participant, such that some faces consistently looked at positive targets (e.g., valid-cue, positive target), whereas other faces never looked at positive targets (invalid-cue, positive target), and so on.”

Comment 4: P.9, line 147. “The faces and the targets in Fig 1 are not the original images that were used in our study, but similar images used for illustrative purposes” – this implies that face images used were line drawings, as in Figure 1. I would remove the word “similar” here and in the legend of Figure 2 on p.11, line 170, or replace it with “schematic” or something similar.

Response 4: We thank for your insightful comments. We have changed the captions according to your comment.

Comment 5: P.11, line 179. Report the results of the test of normality.

Response 5: We thank for your insightful comment. We have added the following texts on captions of page 12, “We performed a Kolmogorov-Smirnov test to check if the distribution of the response times were statistically normal. The test rejected the normality hypothesis for response times of both experiments (Experiment 1: D = 0.15, p < .001; Experiment 2: D = 0.12, p < .001). ”

Comment 6: P.13, line 206 & 218. “reasonably close to significant” is no better than “marginally significant”. It is fine to draw attention to differences or trends, but these should not be oversold. Describe these as non-significant.

Comment 7: P.14, line 220. “we found the effect of face type, F(1, 19) = 3.48, p = .08, ηp2 = .15” – this is not significant, and so post-hoc tests are not justified.

Response 6 and 7: We thank the reviewer for these comments. We have removed some text and rewritten the following sentences on Lines 221-223, “Moreover, there seem to be differences between positive and negative targets, especially for the valid-cue and invalid-cue faces. However, the two-way interaction between face type and target type was not significant, F(1, 19) = 3.41, p = .08, ηp2 = .15.”

Comment 8: 

P.16, line 257. Change “likeability” to “trustworthiness”. See also lines 262, 264, 267, 279, 320, 467-8, & 535.

Response 8: We thank for your insightful comments. We have changed the text.

Comment 9: P.16, line 260. This correlation analysis does not “assess whether attentional orienting was required for increased facial trustworthiness”, as most people would show attentional orienting in response to gaze cues. If trustworthiness is driven by this reorienting you could hypothesis that the magnitude of this attentional reorienting would show a relationship to the trust learning, but this is not the same as testing whether attentional reorienting is required. Gaze following could well be necessary for this effect without there necessarily being a linear relationship between gaze following and trust learning.

Response 9: We welcome this insightful comment. As you have stated, our explanation was insufficient. Accordingly, we have revised the sentences. Moreover, both the Introduction and the Result of Experiment 2 described the prediction of the results. Therefore, we integrated this information in the Introduction on Lines 89-95.

Comment 10: P.27, line 440. “Additional analysis: did evaluation scores influence memory traces of the relationships between face and image?” This title does not match the logic of the analysis described. The described analysis does not test the effect of evaluation on memory, but instead explores whether the results could be explained as a response to the single preceding trial rather than the result of iterative learning. Reword this title to accurately reflect the analysis described.

Response 10: We thank the reviewer for this comment. According to your comment, we have clarified the title as follows (Line 433), “Additional analysis: Can increased facial trustworthiness be explained as a response to a single or an iterative presentation of a face and an image?”

Comment 11: P.30, line 480. The consistency of the target valence may not affect face evaluation, but all faces were consistent in their behaviour (looking towards or away from the target) in Experiment 2. This consistency could win out over the fairly minimal target valence manipulation.

Response 11: We thank the reviewer for this insightful comment. We have added the following text to the Discussion (Line 465-470), “This might be because the gaze direction, either looking towards or away from a target was always consistent within each face identity. The consistency of gaze direction might be more dominant in modulating perceived facial trustworthiness than the consistency of target valence. Therefore, consistent gaze direction might be prioritized in determining facial trustworthiness, which weakens the effect of target valence consistency. It is suggested that future studies examine this possibility.”

Comment 12: P.31, lines 505-508. The lead into describing Kirkham et al is bit weak, and I am not convinced that the weak evidence for learned representations in Experiment 2 fits with the results of that study. Instead, I would rewrite this section, using Kirkham et al. as evidence that participants learn to trust and distrust people based on what they look at and how they respond. A valuable contribution of the current study is that extends this result and shows that, even without an emotional reaction, what people consistently attend can affect how trustworthy they are perceived to be.

Response 12: We thank the reviewer for this comment. According to your comment, we have revised the following sentences in the General Discussion (lines 488-497), “This study demonstrated that the emotional valence of gazed at images affected facial evaluation through gaze cues. The relationship between faces and gazed images, which were learned during the gaze-cue phases, changed facial evaluations. Consistent with this possibility, Kirkham et al. suggested that participants learn to trust or distrust people based on what they look at and how they respond. They demonstrated that faces smiling for a positive scene were evaluated as more trustworthy than faces frowning for a positive scene, suggesting that encoding the face-scene consistency influenced perceived facial trustworthiness. In the light of previous work, our findings suggest that consistent behavioral signals, such as the gaze direction, enhance perceived facial trustworthiness even in the absence of emotional expressions.”

Comment 13: P.32, line 529. “paring” should be “pairing”

Response 13: We thank the reviewer for this comment. We have changed from “paring” to “pairing”.

---

## [Decision Letter · Decision Letter 3]

26 May 2020

PONE-D-19-14867R3

Affective evaluation of images influences personality judgments through gaze perception

PLOS ONE

Dear Dr. Shirai,

Thank you once again for submitting your manuscript to PLOS ONE.

Both the reviewers and I believe that your paper has much improved since the last revision; however, there are a few minor concerns that need to be addressed before your paper is accepted for publication. In particular, there are sections of the Introduction and Results/Discussion that need to be further clarified, including your discussion of the Bayliss and Tipper (2006) paper, discussion of gaze cueing in the positive target condition, and how your power analysis was conducted.

We look forward to receiving your revised manuscript.

Kind regards,

Veronica Whitford, Ph.D.

Academic Editor

PLOS ONE

Reviewers' comments:

Reviewer's Responses to Questions

**Comments to the Author**

1. If the authors have adequately addressed your comments raised in a previous round of review and you feel that this manuscript is now acceptable for publication, you may indicate that here to bypass the “Comments to the Author” section, enter your conflict of interest statement in the “Confidential to Editor” section, and submit your "Accept" recommendation.

Reviewer #1: (No Response)

Reviewer #3: (No Response)

2. Is the manuscript technically sound, and do the data support the conclusions?

Reviewer #1: Yes

Reviewer #3: Yes

3. Has the statistical analysis been performed appropriately and rigorously? 

Reviewer #1: Yes

Reviewer #3: Yes

4. Have the authors made all data underlying the findings in their manuscript fully available?

Reviewer #1: Yes

Reviewer #3: Yes

5. Is the manuscript presented in an intelligible fashion and written in standard English?

Reviewer #1: Yes

Reviewer #3: Yes

6. Review Comments to the Author

Reviewer #1: 1) change to lines 88-95: Upon first reading this section, I got the impression that the authors were trying to say that some research shows a link between perceived trustworthiness and gaze cueing, while others do not. If that is the case, then the Bayliss & Tipper (2006) paper should not be listed as evidence of the former, but rather of the latter. I think what I’m confused by is the use of “attentional facilitation” and “[magnitude] of [gaze] cueing effects”, and how/whether they differ. If the authors mean for them to be the same, then using consistent wording would get that across more clearly.

After looking at the references used to support the two sides, my second impression is that the authors are trying to delineate whether gaze preference leads to liking certain objects more, or liking (trusting) certain faces more? So the directionality between gaze conferring special status on objects, versus gaze behaviours ‘rebounding’ onto that individual to guide trustworthy judgments of said face.

In either case, I believe the authors need to be more explicit about what they mean here.

As an aside, I suggest using the term “magnitude” rather than “degree” when discussing modulations of the cueing effect throughout the text.

2) invalid-cue face prediction: I am satisfied with this change

4) Design location: I am satisfied with this change

5) lines 134-136: I am satisfied with this change

6) lines 155-157: I am satisfied with this change

7) Figure 2 caption: this is much improved. I would suggest removing the word “pairs” from “there were 6 pairs of faces” and “there were 8 pairs of faces” – this reads as though you only had 12 faces in E1 and 16 faces in E2, which is not true.

8) Gaze cueing in positive target condition: I would argue that the authors can only draw conclusions about the magnitude of the gaze cueing effect if the ANOVA contrasted just valid and invalid trials. While some of the early gaze cueing literature used a neutral condition to calculate costs and benefits (e.g., Friesen & Kingstone, 1998), this is no longer the norm; rather, a direct contrast between valid and invalid trials is conducted. If the authors run an ANOVA with face type (valid-cue, invalid-cue) and target type (negative, positive), do they see a main effect of face type that does not interact with target type. This analysis is more in-line with the vast majority of analyses in this field, thus I believe it would be warranted to include it. Otherwise, one could argue that the much slower RTs in the no-cue condition were driving the ‘cueing effect’ the authors claim is present.

9) lines 220-222: the authors should reword this section slightly. “While numerically there appears to be a difference between the positive and negative targets, the two-way interaction between …”

10 & 11) I am satisfied with these changes

12 & 13) Bayes (lines 400-408): The authors should remove the writing between lines 400 and 405, beginning “Moreover, BF10...” and concluding “(BF10 = 1.88)”. The highlighted new section provides the Bayes analyses without overselling them or having the findings repeated twice.

15) E2 discussion: I am satisfied with this change

Reviewer #3: This paper is much improved and the authors have addressed most of my comments from the previous round adequately.

The only point I am unsure about (which I did not notice in the last revision) is the power calculation for Experiment 1 where the authors use an intraclass correlation coefficient of zero. As far as I understand, the ICC in this case refers to the similarity between values from the same participant. An ICC of zero therefore predicts no individual variation in the sample (e.g. that individuals will not give overall higher ratings or overall lower ratings than others ). This does not seem a plausible prediction to me, but as I do not have the statistical expertise to say how such an assumption affects the sample size estimate I will leave this to the authors and AE.

The rest of my points are all suggestions of ways to further clarify and simplify the text that I urge the authors to consider.

- “We predicted that the emotional valence of images would be more transferred to the gazer’s personality when the sender’s gaze direction matched the location of emotional images” – This is unclear, and ‘more transferred’ is an odd way of phrasing. What is wrong with just saying, “emotional valence of images would affect personality judgements only when the face looked toward the image rather than away from it”?

- Once again, I caution the authors to be mindful of confusing likeability and trustworthiness (e.g. p.6 lines 91-93)

- P.9-10 lines 146-156: This covers predictions about the third cueing condition that should have been spelt out in the introduction

- Change the labels on Figs 1 and 2, as currently Figure 2 appears first in the manuscript.

- P.14 line 231-232: “A BF10 greater than 3.0 provides moderate support for the null hypothesis” – This is incorrect. This should be either “BF01”, or “alternative hypothesis”

- P.14 line 233-235: The authors have just explained how BF10 works, so there is no need to repeat this in such confusing terms. Spell out how the predictions will translate to results in clear, simple terms (e.g. “Under the hypothesis that the valence of gazed images affects facial evaluation, we expected to find strong evidence (BF10 > 10) of a difference in ratings of valid faces between positive and negative targets.”)

- P.15 line 242: Similarly, “This finding suggests the validity of models supporting the hypothesis that…” is overly complicated and difficult to parse. It is much simpler to just say, “This finding supports the hypothesis that…” It is worth going through the Results section (the sections with Bayesian analysis in particular) and simplifying the sentences wherever possible to achieve maximum clarity.

- P.20: Why is the result of the power analysis different in Experiment 2 if it used the same criteria as that of Experiment 1? If the criteria were different (i.e. the effect size was changed to match an observed effect size from Experiment 1) then this needs to be reported.

- P.23 line 378: “target position cues” should read “valence-target and position cues”, no?

- P.23 footnote: “differences […] were not significant and that in both cases the faces gaze toward the negative [t] and positive images, [t]” – The wording of this suggests that these stats tested which way the faces were looking, rather than changes in evaluations to different types of faces. I suggest changing this to, “differences […] were not significant, both in cases where…”

- P.27: “The BF10 supporting the hypothesis that… was slightly higher than BF01, which supported the null” – Again, this is unclear because it suggests that the results support the null when in fact it is the opposite. Use “support” exclusively to refer to the current results and simplify the description of the results overall to avoid such misinterpretations. Also, bear in mind that there is little need to report both BF01 and BF10 at the same time – these values are the inverse of each other, and so if BF10 is >1 then that is evidence supporting the alternative, and if it is <1 then that is evidence supporting the null (although it may not be worth interpreting if it is so close to 1). Just reporting one of these may help to simplify the reporting of these tests substantially.

- P.29 line 473: “to clarify whether the emotional valence of a gaze target affected the impression of the gazers’.” should read, “gazer’s trustworthiness” or something similar.

7. PLOS authors have the option to publish the peer review history of their article (what does this mean?). If published, this will include your full peer review and any attached files.

Reviewer #1: No

Reviewer #3: No

---

## [Author Response · Author response to Decision Letter 3]

28 Jul 2020

RESPONSE TO REVIEWER 1:

Reference: PONE-D-19-14867R3

Resubmission

“Affective evaluation of images influences personality judgments through gaze perception”

We highly appreciate you for taking the time to make valuable comments on the above entitled manuscript despite your busy schedule. We have revised the manuscript according to your comments. Our responses to each reviewers’ comments are listed below.

Comment 1: change to lines 88-95: Upon first reading this section, I got the impression that the authors were trying to say that some research shows a link between perceived trustworthiness and gaze cueing, while others do not. If that is the case, then the Bayliss & Tipper (2006) paper should not be listed as evidence of the former, but rather of the latter. I think what I’m confused by is the use of “attentional facilitation” and “[magnitude] of [gaze] cueing effects”, and how/whether they differ. If the authors mean for them to be the same, then using consistent wording would get that across more clearly.

After looking at the references used to support the two sides, my second impression is that the authors are trying to delineate whether gaze preference leads to liking certain objects more, or liking (trusting) certain faces more? So the directionality between gaze conferring special status on objects, versus gaze behaviours ‘rebounding’ onto that individual to guide trustworthy judgments of said face.

In either case, I believe the authors need to be more explicit about what they mean here.

As an aside, I suggest using the term “magnitude” rather than “degree” when discussing modulations of the cueing effect throughout the text.

Response 1: Thank you for this insightful comment. 

As you have stated, this sentence was unclear. After considering all the reviewers’ comments comprehensively, we considered that the section on this correlation analysis might confuse the readers’ understanding of the main results of our study because the hypothesis of the correlation analysis was developed from a different perspective from the main body of this study. 

Moreover, as you have pointed out, we also considered it difficult to draw clear conclusions and develop a meaningful discussion based only on this correlation analysis. Therefore, we removed the description of the correlation analysis from the revised manuscript. We believe that this change does not significantly influence our study’s primary results and hope that this change would contribute to a better understanding of our study.

Comment 2: Figure 2 caption: this is much improved. I would suggest removing the word “pairs” from “there were 6 pairs of faces” and “there were 8 pairs of faces” – this reads as though you only had 12 faces in E1 and 16 faces in E2, which is not true.

Response 2: 

Thank you for this comment. We have removed the word “pairs” according to your comment. 

Comment 3: Gaze cueing in positive target condition: I would argue that the authors can only draw conclusions about the magnitude of the gaze cueing effect if the ANOVA contrasted just valid and invalid trials. While some of the early gaze cueing literature used a neutral condition to calculate costs and benefits (e.g., Friesen & Kingstone, 1998), this is no longer the norm; rather, a direct contrast between valid and invalid trials is conducted. If the authors run an ANOVA with face type (valid-cue, invalid-cue) and target type (negative, positive), do they see a main effect of face type that does not interact with target type. This analysis is more in-line with the vast majority of analyses in this field, thus I believe it would be warranted to include it. Otherwise, one could argue that the much slower RTs in the no-cue condition were driving the ‘cueing effect’ the authors claim is present.

Response 3: We welcome this insightful comment. As you have stated, we also consider conducting an analysis that is similar to the vast majority if analyses in this field would help the readers better interpret our results. Therefore, we have included the following sentences in Lines 215-238, based on your comment, “We conducted an ANOVA for the no-cue condition described above. However, nearly all gaze cueing studies examining the effect of gaze cues have directly contrasted response times between valid- and invalid-cue conditions. Therefore, we additionally conducted an ANOVA with face type (valid-cue and invalid-cue) and target type (negative and positive), similar to many previous studies. Results indicated that the main effect of face type (F(1, 19) = 8.01, p = .01, ηp2 = .30) and its interaction with target type were significant (F(1, 19) = 4.42, p = .05, ηp2 = .19). Multiple comparisons showed that the response times were faster for valid than for invalid cue trials when the target was negative (F(1, 19) = 11.29, p < .01, ηp2 = .37), which was not the case when the target type was positive (F(1, 19) = 1.75, p = .20, ηp2 = .08). Moreover, the response times for valid cue trials were faster when the target type was negative than when the target type was positive (F(1, 19) = 6.19, p < .05, ηp2 = .25). There were no significant differences in response times between positive and negative targets for invalid cue trials (F(1, 19) = 0.64, p = .43, ηp2 = .03). Moreover, the main effect of target type was not significant, F(1, 19) = 0.80, p = .38, ηp2 = .04.”

Comment 4: lines 220-222: the authors should reword this section slightly. “While numerically there appears to be a difference between the positive and negative targets, the two-way interaction between …”

Response 4: Thank you for your comment. According to your comments, we have changed the description of this point (Lines 285-287) as follows, “While numerically there appears to be a difference between the positive and negative targets, the two-way interaction between face type and target type was not significant, F(1, 19) = 3.41, p = .08, ηp2 = .15.”

Comment 5: Bayes (lines 400-408): The authors should remove the writing between lines 400 and 405, beginning “Moreover, BF10...” and concluding “(BF10 = 1.88)”. The highlighted new section provides the Bayes analyses without overselling them or having the findings repeated twice.

Response 5: We thank the reviewer for this valuable comment. We removed these sentences in the revised manuscript according to your comment. Moreover, we changed the description of the Bayes analysis to exclude repetitions (Lines 514-525).

RESPONSE TO REVIEWER 2:

Reference: PONE-D-19-14867R3

Resubmission

“Affective evaluation of images influences personality judgments through gaze perception”

We highly appreciate you for taking the time to make valuable comments on the above entitled manuscript despite your busy schedule. We have revised the manuscript according to your comments. Our responses to each reviewers’ comments are listed below.

Comment 1: The only point I am unsure about (which I did not notice in the last revision) is the power calculation for Experiment 1 where the authors use an intraclass correlation coefficient of zero. As far as I understand, the ICC in this case refers to the similarity between values from the same participant. An ICC of zero therefore predicts no individual variation in the sample (e.g. that individuals will not give overall higher ratings or overall lower ratings than others ). This does not seem a plausible prediction to me, but as I do not have the statistical expertise to say how such an assumption affects the sample size estimate I will leave this to the authors and AE.

Response 1: Thank you for this insightful comment. We agree with your point. There is some debate on ICC settings in situations in which the ICC cannot be known in advance, and there are recommendations for setting it to 0.5 or 0. In this study, we used 0. We believe that the ICC setting did not significantly influence the results of our study. 

Comment 2: “We predicted that the emotional valence of images would be more transferred to the gazer’s personality when the sender’s gaze direction matched the location of emotional images” – This is unclear, and ‘more transferred’ is an odd way of phrasing. What is wrong with just saying, “emotional valence of images would affect personality judgements only when the face looked toward the image rather than away from it”?

Response 2: Thank you for your insightful comments. Based on your comment, we revised this sentence in the revised manuscript as follows, “We predicted that the emotional valence of images would affect the personality judgments only when the face looked toward the image rather than away from it.” (in lines 83-86)

Comment 3: Once again, I caution the authors to be mindful of confusing likeability and trustworthiness (e.g. p.6 Lines 91-93)

Response 3: We thank the reviewer for this comment. We have revised manuscript with attention to the use of words, likability and trustworthiness.

Comment 4: P.9-10 Lines 146-156: This covers predictions about the third cueing condition that should have been spelt out in the introduction

Response 4: We thank you for your insightful comments. We have moved these sentences to the introduction of the revised manuscript according to your comment (Lines 90-98). 

Comment 5: Change the labels on Figs 1 and 2, as currently Figure 2 appears first in the manuscript.

Response 5: We thank you for your insightful comment. We changed the labels on Figures 1 and 2 of the revised manuscript.

Comment 6: P.14 line 231-232: “A BF10 greater than 3.0 provides moderate support for the null hypothesis” – This is incorrect. This should be either “BF01”, or “alternative hypothesis” 

Comment 7: P.14 line 233-235: The authors have just explained how BF10 works, so there is no need to repeat this in such confusing terms. Spell out how the predictions will translate to results in clear, simple terms (e.g. “Under the hypothesis that the valence of gazed images affects facial evaluation, we expected to find strong evidence (BF10 > 10) of a difference in ratings of valid faces between positive and negative targets.”)

Comment 8: P.15 line 242: Similarly, “This finding suggests the validity of models supporting the hypothesis that…” is overly complicated and difficult to parse. It is much simpler to just say, “This finding supports the hypothesis that…” It is worth going through the Results section (the sections with Bayesian analysis in particular) and simplifying the sentences wherever possible to achieve maximum clarity.

Comment 12: P.27: “The BF10 supporting the hypothesis that… was slightly higher than BF01, which supported the null” – Again, this is unclear because it suggests that the results support the null when in fact it is the opposite. Use “support” exclusively to refer to the current results and simplify the description of the results overall to avoid such misinterpretations. Also, bear in mind that there is little need to report both BF01 and BF10 at the same time – these values are the inverse of each other, and so if BF10 is >1 then that is evidence supporting the alternative, and if it is <1 then that is evidence supporting the null (although it may not be worth interpreting if it is so close to 1). Just reporting one of these may help to simplify the reporting of these tests substantially.

Response 6, 7, 8, and 12: We thank you for your insightful comment. It is correct that “A BF10 greater than 3.0 provides moderate support for the null hypothesis”. We have fixed these mistakes. Moreover, according to your comments, we also simplified the reporting of BF results to make it easier to understand (Lines 301-311, 518-531, 547-555).

Comment 9: P.20: Why is the result of the power analysis different in Experiment 2 if it used the same criteria as that of Experiment 1? If the criteria were different (i.e. the effect size was changed to match an observed effect size from Experiment 1) then this needs to be reported.

Response 9: We thank you for your insightful comment. The results of the power analysis were different because the factorial design was different between Experiment 1 (3 × 2 factorial design) and Experiment 2 (2 × 4 factorial design). 

Comment 10: P.23 line 378: “target position cues” should read “valence-target and position cues”, no?

Response 10: We thank you for your insightful comment. You are right. We have changed this sentence to “valence-target and position cues” (Line 507).

Comment 11: P.23 footnote: “differences […] were not significant and that in both cases the faces gaze toward the negative [t] and positive images, [t]” – The wording of this suggests that these stats tested which way the faces were looking, rather than changes in evaluations to different types of faces. I suggest changing this to, “differences […] were not significant, both in cases where…”

Response 11: We thank the reviewer for this insightful comment. We changed this sentence in the revised manuscript based on your comment as follows, “The results showed that differences between the number of evaluations were neither significant in cases in which faces gazed toward negative, t(45.27) = -0.49, p = .62, nor positive images, t(45.94) = 0.11, p = .92, suggesting that the number of evaluations did not affect facial evaluations in Experiment 2.” (P. 22 footnote)

Comment 13: P.29 line 473: “to clarify whether the emotional valence of a gaze target affected the impression of the gazers’.” should read, “gazer’s trustworthiness” or something similar.

Response 13: We thank the reviewer for this comment. We changed this sentence in the revised manuscript according to your comment as follows, “The present study was designed to clarify whether the emotional valence of a gazed target affected the gazer’s trustworthiness”. (Line 574)

---

## [Decision Letter · Decision Letter 4]

4 Sep 2020

PONE-D-19-14867R4

Affective evaluation of images influences personality judgments through gaze perception

PLOS ONE

Dear Dr. Shirai,

Thank you once again for considering PLOS ONE as an outlet for your research. The reviewers and I both agree that your manuscript has significantly improved through the revision process and that it will make an important contribution to the empirical literature. Although Reviewer 1 recommends publication, Reviewer 2 has a residual concern regarding your power calculation that should be addressed. Therefore, I am inviting you to address this lingering issue in a revised version of the manuscript.

Given that few revisions are required this round, could you please submit your revised manuscript by October 5, 2020? If you will need more time than this to complete your revisions, please reply to this message or contact the journal office at plosone@plos.org. Please include the following items when submitting your revised manuscript:

We look forward to receiving your revised manuscript.

Kind regards,

Veronica Whitford, Ph.D.

Academic Editor

PLOS ONE

Reviewers' comments:

Reviewer's Responses to Questions

**Comments to the Author**

1. If the authors have adequately addressed your comments raised in a previous round of review and you feel that this manuscript is now acceptable for publication, you may indicate that here to bypass the “Comments to the Author” section, enter your conflict of interest statement in the “Confidential to Editor” section, and submit your "Accept" recommendation.

Reviewer #1: (No Response)

Reviewer #3: (No Response)

2. Is the manuscript technically sound, and do the data support the conclusions?

Reviewer #1: Yes

Reviewer #3: Yes

3. Has the statistical analysis been performed appropriately and rigorously? 

Reviewer #1: Yes

Reviewer #3: Yes

4. Have the authors made all data underlying the findings in their manuscript fully available?

Reviewer #1: Yes

Reviewer #3: Yes

5. Is the manuscript presented in an intelligible fashion and written in standard English?

Reviewer #1: Yes

Reviewer #3: Yes

6. Review Comments to the Author

Reviewer #1: 1) change to lines 88-95: this reads much better! One tiny change I would suggest is to use the term “target” instead of “image”: We predicted that the emotional valence of images would affect personality judgments only when the face looked toward the TARGET rather than away from it.” – the authors should make this change in the subsequent sentence as well, for clarity.

2) Figure captions: I believe the authors swapped figures 1 and 2, however didn’t rename the corresponding figure captions. Further, while they took out the word “pairs”, the did not take out the word “of” that came after the word “pairs”. Please proofread this caption to remove the errors.

3) Additional ANOVA: Thank you for running the additional analysis. Please provide a few references to support the sentence stating that “nearly all gaze cueing studies…invalid-cue conditions.”

4) lines 220-222: I am satisfied with this change

5) Bayes: I am satisfied with this change

Reviewer #3: The authors have addressed most of my comments on the previous version of this manuscript to my satisfaction. The paper now is very well written, has clearly described methods and results, and bases clear conclusions on the basis of this evidence. I think that the decision to drop the correlation analysis from the main text was a good one as it was clearly motivated by different hypotheses than the authors set out to test, although I think it may be preferable to report this analysis in the Supplemental Material than to leave it out completely (particularly as the data are still included in the S2 Appendix).

I would like to say – before I start what will seem like a lengthy review – that I am very pleased with the state of this manuscript and think it should be published. It addresses an interesting question, it makes a good contribution to the literature, and I very much appreciate the work that the authors have put in over this review process in response to my and the other reviewer’s comments, which has made the whole study much clearer and the manuscript much stronger.

I have one outstanding major issue with the power analysis. On the basis of the authors’ response to my comment about the ICC I became interested in how this coefficient would change the results of the sample size calculation and so decided to download G*Power and try it out. Unfortunately, I am unable to replicate the sample size estimates on the basis of the values reported in the manuscript.

It is possible that I overlooked a value or have misunderstood something that the authors know about the logic of G*Power, but I think it is also possible that the sample size criteria in this study were based on erroneous calculations. For example, G*Power is unable to handle designs with more than one within-subject factor (see https://www.researchgate.net/post/How_compute_a_repeated_measure_power_analysis_in_Gpower for a description and the workaround solution by David Morse that I used when trying to replicate the analysis). The only way to know is to allow readers to fully replicate the power analyses.

I have listed the G*Power settings that should be reported, with the values I used for my attempt at replication based on the 2x3 within-subjects interaction in Experiment 1 (ANOVA: Repeated measures, within factors; f=0.25; alpha=0.05; 1-beta=0.80; no. groups=1; no. measurements=3; corr. among rep measures=0.0; nonsphericity correction=1 – estimated sample size: 53).

I also advise the authors not to use G*Power again to estimate power for future studies using a within-subjects factorial design. Simulation-based approaches (such as using the Superpower app: http://shiny.ieis.tue.nl/anova_power) are generally better suited for this.

There are just a couple of minor points that the authors should correct before submitting their final version of the manuscript.

p.5, line 70 ‘module’ should be ‘modulate’

The figure legends in the text are still incorrect – ‘Fig 2’ on p.9 is the first figure, for example, and the legend refers to the figure that has been uploaded as Figure 2, while Figure 1 does not appear until p.11. It is not enough to swap the labels around – the authors should take care that the figures are presented, labelled, and referred to appropriately throughout the document.

p.23, lines 378-381: “Results showed that the BF10, in the comparison between positive and negative valence-target conditions, was 6.82, which moderately supports the alternative hypothesis. This result supported our contention that faces always looking at positive images are evaluated as more trustworthy than faces always looking at negative images.” – Specify that this contrast is for valid faces (i.e. “the comparison of ratings for valid faces…”)

p.25 The additional analysis is not reproducible using the data available in the Supplemental Material. This requires at least having the ratings for valid-random faces broken down by the target they looked at on the previous trial.

7. PLOS authors have the option to publish the peer review history of their article (what does this mean?). If published, this will include your full peer review and any attached files.

Reviewer #1: No

Reviewer #3: No

---

## [Author Response · Author response to Decision Letter 4]

25 Sep 2020

RESPONSE TO REVIEWER 1:

Reference: PONE-D-19-14867R4

Resubmission

“Affective evaluation of images influences personality judgments through gaze perception”

We highly appreciate you for taking the time off your busy schedule to make valuable comments on the above entitled manuscript. We have revised the manuscript according to your comments. Our responses to each reviewers’ comments are listed below.

Comment 1: change to lines 88-95: this reads much better! One tiny change I would suggest is to use the term “target” instead of “image”: We predicted that the emotional valence of images would affect personality judgments only when the face looked toward the TARGET rather than away from it.” – the authors should make this change in the subsequent sentence as well, for clarity.

Response 1: We have changed from “image” to “target,” according to your comment (line 85).

Comment 2: Figure captions: I believe the authors swapped figures 1 and 2, however didn’t rename the corresponding figure captions. Further, while they took out the word “pairs”, the did not take out the word “of” that came after the word “pairs”. Please proofread this caption to remove the errors.

Response 2: Thank you for this comment. 

We carefully proofread the manuscript and removed the word “of ” according to your comment (lines 168, 169). 

Comment 3: Additional ANOVA: Thank you for running the additional analysis. Please provide a few references to support the sentence stating that “nearly all gaze cueing studies…invalid-cue conditions.”

Response 3: We thank the reviewer for this comment.

We have provided references to support this sentence (line 217).

RESPONSE TO REVIEWER 3:

Reference: PONE-D-19-14867R4

Resubmission

“Affective evaluation of images influences personality judgments through gaze perception”

We highly appreciate you for taking the time off your busy schedule to make valuable comments on the above-entitled manuscript. We have revised the manuscript according to your comments. Our responses to each reviewers’ comments are listed below.

Comment 1: I think that the decision to drop the correlation analysis from the main text was a good one as it was clearly motivated by different hypotheses than the authors set out to test, although I think it may be preferable to report this analysis in the Supplemental Material than to leave it out completely (particularly as the data are still included in the S2 Appendix).

Response 1: Thank you for this insightful comment. We considered that the correlation analysis did not provide the readers with a clear and specific conclusion. Therefore, we decided to remove information about the correlation analysis from the Supplemental Material and S2 Appendix.

Comment 2: It is possible that I overlooked a value or have misunderstood something that the authors know about the logic of G*Power, but I think it is also possible that the sample size criteria in this study were based on erroneous calculations. For example, G*Power is unable to handle designs with more than one within-subject factor (see https://www.researchgate.net/post/How_compute_a_repeated_measure_power_analysis_in_Gpower for a description and the workaround solution by David Morse that I used when trying to replicate the analysis). The only way to know is to allow readers to fully replicate the power analyses.

I have listed the G*Power settings that should be reported, with the values I used for my attempt at replication based on the 2x3 within-subjects interaction in Experiment 1 (ANOVA: Repeated measures, within factors; f=0.25; alpha=0.05; 1-beta=0.80; no. groups=1; no. measurements=3; corr. among rep measures=0.0; nonsphericity correction=1 – estimated sample size: 53).

I also advise the authors not to use G*Power again to estimate power for future studies using a within-subjects factorial design. Simulation-based approaches (such as using the Superpower app: http://shiny.ieis.tue.nl/anova_power) are generally better suited for this.

Response 2: Thank you for your insightful comments. 

As you have stated, we found that the G*Power settings in this study might be odd. 

We have listed the G*Power settings used in Experiment 1 of our study (ANOVA: Repeated measures, within factors; f = 0.25; alpha = 0.05; 1-beta = 0.80; number of groups = 1; number of measurements = 6; corr. among rep measures = 0.0; nonsphericity correction = 0.2 (1/ (number of measurements-1)); estimated sample size: 107/6 (number of measurements) = 17.83).

Moreover, using G*power may not have been the best choice among the various power analysis methods. We highly appreciate the reviewer for providing information on simulation-based power analysis approaches. We would like to use this method in the future studies. We calculated the power in our study by using the power analysis that you recommended. As a result, the power of the 2x3 within-subjects interactions was 86.42 % when the sample size was 20. 

Therefore, we believe that the power was not too small. 

Since the current study used the G*power for the sample size calculation, we believe that information about the G*power should be included in this paper. 

Thus, we have added and rewritten the explanation of G*power's analysis to avoid providing misleading information to readers based on your comments as follows “We conducted an advance power analysis (G*power; [16, 17]) in to determine the minimum sample size by adopting the following settings: a medium effect size (f) of 0.25, a significance level of 5%, 1 group, 6 measurements, a correction for nonsphericity of 0.2, and an intra-class correlation coefficient of zero. It was decided that a minimum of 18 participants was needed to achieve a power level of 0.80.” (lines 103-106, 325-326).

Comment 3: p.5, line 70 ‘module’ should be ‘modulate’

Response 3: We thank the reviewer for this comment. We have changed ‘module’ to ‘modulate’ (line 70).

Comment 4: The figure legends in the text are still incorrect – ‘Fig 2’ on p.9 is the first figure, for example, and the legend refers to the figure that has been uploaded as Figure 2, while Figure 1 does not appear until p.11. It is not enough to swap the labels around – the authors should take care that the figures are presented, labelled, and referred to appropriately throughout the document.

Response 4: We apologize for making you point out this mistake repeatedly. We thank you for your repeated correction. We have carefully revised and changed the legends of the Figures according to your comment. 

Comment 5: p.23, lines 378-381: “Results showed that the BF10, in the comparison between positive and negative valence-target conditions, was 6.82, which moderately supports the alternative hypothesis. This result supported our contention that faces always looking at positive images are evaluated as more trustworthy than faces always looking at negative images.” – Specify that this contrast is for valid faces (i.e. “the comparison of ratings for valid faces…”)

Response 5: We thank you for your insightful comment. We have added an explanation about the condition to clarify this point as follows, “Results indicated that BF10 for comparing positive and negative valence-target conditions for valid faces was 6.82, which moderately supported the alternative hypothesis.” (lines 396-397).

Comment 6: p.25 The additional analysis is not reproducible using the data available in the Supplemental Material. This requires at least having the ratings for valid-random faces broken down by the target they looked at on the previous trial. 

Response 6: We thank you for your insightful comment. We included data regarding the additional analysis in the Supplemental Material, according to your comment.

---

## [Editor Report · Decision Letter 5]

14 Oct 2020

Affective evaluation of images influences personality judgments through gaze perception

PONE-D-19-14867R5

Dear Dr. Shirai,

We’re pleased to inform you that your manuscript has been judged scientifically suitable for publication and will be formally accepted for publication once it meets all outstanding technical requirements.

Kind regards,

Veronica Whitford, Ph.D.

Academic Editor

PLOS ONE

---

## [Editor Report · Acceptance letter]

20 Oct 2020

PONE-D-19-14867R5 

Affective evaluation of images influences personality judgments through gaze perception 

Dear Dr. Shirai:

I'm pleased to inform you that your manuscript has been deemed suitable for publication in PLOS ONE. Congratulations! Your manuscript is now with our production department. 

Kind regards, 

on behalf of

Dr. Veronica Whitford 

Academic Editor

PLOS ONE